# The vertical structure of the troposphere and its connection to the surface mass balance of Flade Isblink in Northeast Greenland

Jonathan Fipper[1,2], Jakob Abermann[1], Ingo Sasgen[2,3], Henrik Skov[4], Lise Lotte Sørensen[4], Wolfgang Schöner[1]

[1]Department of Geography and Regional Science, University of Graz, Graz, 8010, Austria
[2]Glaciology Section, Division of Geosciences, Alfred Wegener Institute Helmholtz Centre for Polar and Marine Research, Bremerhaven, 27568, Germany
[3]Institute of Geography, University of Augsburg, Alter Postweg 118, 86159 Augsburg, Germany
[4]Department of Environmental Science, iClimate, ARC, Aarhus University, Roskilde, Denmark

*Correspondence to*: Jonathan Fipper (jonathan@fipper.de)

**Abstract.** Glaciers and ice caps (GIC) north of 79° N in Greenland contributed around 60 % to the total mass loss of all GIC in Greenland between 2018 and 2021, driven largely by surface melt in response to rising temperatures. Vertical temperature structures in the lower atmosphere modulate the surface energy exchanges and are an important factor in governing surface melt. Despite this importance, few in situ observations are available. We measured 130 vertical air temperature profiles up to 500 m above ground using uncrewed aerial vehicles (UAVs) over different surface types around Villum Research Station (VRS) in Northeast Greenland. VRS is 5 km west of Flade Isblink ice cap (FIIC), the largest peripheral ice mass in Greenland. We find a robust agreement between the UAV temperature profiles and the ones of the Copernicus Arctic Regional Reanalysis (CARRA) data set (mean absolute difference of 1 °C; r = 0.59), which allows us to use CARRA for a detailed process study. Using daily CARRA data for June, July and August from 1991 to 2024, we find that surface properties significantly ($\alpha = 0.01$) control air temperature variability up to ~100 m above ground. K-means clustering of vertical temperature gradients above 100 m above ground reveals that the profiles reflect distinct large-scale synoptic conditions. We assess the influence of the synoptic conditions on the surface mass balance (SMB) of FIIC using output from the Modèle Atmosphérique Régional (MAR). Overall, mass loss of ~21 Gt occurred since 2015, driven by summer air temperatures under all synoptic conditions. The most extreme melt season with a SMB of -0.8 m water equivalent and an equilibrium line altitude estimated 467 m above average occurred in 2023, associated with frequent synoptic conditions favourable for melt.

## 1 Introduction

Anthropogenic climate change is unequivocally warming the planet (IPCC, 2021). Nowhere is this more pronounced than in the Arctic, which is warming nearly four times faster than the global average during recent decades, a phenomenon known as Arctic amplification (Box et al., 2019; Rantanen et al., 2022). In Greenland, both the ice sheet and peripheral glaciers and ice

caps (GIC) have been losing mass at an accelerating rate over the last decades in response to climate warming (Hanna et al., 2021; Khan et al., 2022). Although GIC account for only ~4 % of Greenland's glacier covered area, they currently contribute about 11 % to the total ice mass loss (Khan et al., 2022; The GlaMBIE Team, 2025; Carrivick et al., 2023). Especially, the GIC north of ~79° N have shown increased mass loss over the past years, contributing with around 60 % of the total mass loss of all GIC from 2018 to 2021 (Khan et al., 2022; Noël et al., 2017; Carrivick et al., 2023).

The mass loss of the GIC in Greenland is primarily driven by surface melt, which is controlled by a complex interplay of surface energy exchanges tightly coupled with surface albedo and atmospheric conditions in the lower troposphere (Bollen et al., 2023; Noël et al., 2017; Gardner et al., 2013; Shahi et al., 2020). Vertical tropospheric temperature structures significantly influence the surface energy balance, and thus surface melt (Chutko and Lamoureux, 2009; Mernild and Liston, 2010; Hansche et al., 2023). However, due to the limited availability of in situ observations and the complexity of surface coverage types and feedback mechanisms, these vertical structures and their role in the increased mass loss of North Greenland's GIC remain poorly understood (Schuster et al., 2021; Khan et al., 2022; Mernild and Liston, 2010).

Studying vertical temperature structures in Greenland is challenging. Radiosonde observations are sparse and typically offer only low vertical resolution, in particular near the surface (Gilson et al., 2018; Hansche et al., 2023) and so are studies using tethered balloons (Egerer et al., 2023). Studies based on reanalysis products or modelling have been conducted, but their reliability is difficult to assess in the high Arctic with sparse observations (Mernild and Liston, 2010; Ruman et al., 2022; Shahi et al., 2020). In recent years, uncrewed aerial vehicles (UAVs) have emerged as valuable tools to observe atmospheric profiles with high spatial and temporal resolution (Hansche et al., 2023; Kral et al., 2018; Barbieri et al., 2019).

This study combines UAV-based soundings of the lower troposphere at Villum Research Station (VRS), taken over different surface types (sea water, glacier, tundra, lake), with regional climate model output and reanalysis data to investigate vertical temperature structures and their connection to synoptic circulation patterns, as well as the surface mass balance (SMB) of the peripheral Flade Isblink Ice Cap (FIIC). To meet this overarching aim, we pursue the following objectives: (1) validate reanalysis data by UAV-based atmospheric profiles; (2) identify the drivers of vertical temperature structures, focusing on the role of surface properties and the large scale atmospheric circulation; (3) quantify the SMB of FIIC and its key drivers over the last decades; and (4) discuss the influence of vertical temperature structures on the ice cap's SMB.

## 2    Methods

### 2.1 Study Area

The VRS is located next to the Danish military outpost Station Nord at around 81° N and 16° W in Crown Prince Christian Land, Northeast Greenland (Fig. 1). This region experiences mean annual air temperatures of -15.1 °C and thick, long-lasting snow cover typically exceeding 1 m (Strand et al., 2022; van der Schot et al., 2024). The warmest month is July, with a mean air temperature of 4 °C while the coldest month, March, reaches -26 °C (Gryning et al., 2021).

FIIC is located adjacent to VRS and with ∼7,500 km² the largest peripheral ice mass in Greenland (Möller et al., 2022). The ice cap has several marine terminating outlet glaciers, some of which have exhibited surge behaviour in recent decades (Palmer et al., 2010; Möller et al., 2022). For 1990 to 2010, the ice cap's mass changes were estimated to be in balance (Bolch et al., 2013; Rinne et al., 2011; Gardner et al., 2013). More recently, signs of increased mass loss have been observed, consistent with trends across other GIC in North Greenland (Khan et al., 2022).

The relatively flat terrain around VRS and its proximity to FIIC and various surface types make the site ideal for UAV-based atmospheric profiling. The station's logistical accessibility and its continuous atmospheric monitoring further enhance its suitability for studying temperature structures and their connection to the SMB of FIIC. By using the long-term atmospheric observations at VRS, Gryning et al. (2023) demonstrated that the atmospheric boundary layer at the site is shallow.

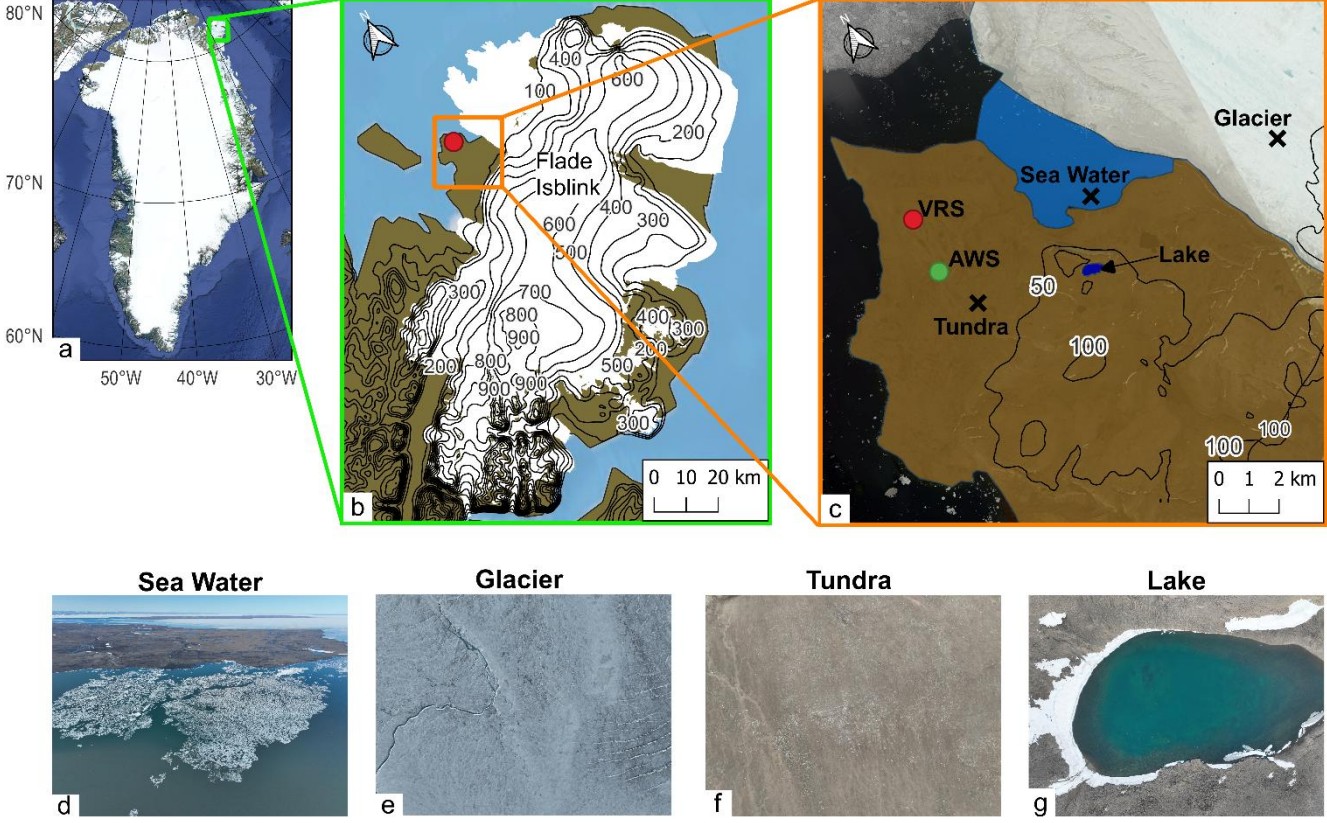

**Figure 1: Overview of the study area in Northeast Greenland (a-c). The red dot in panels (b) and (c) marks the location of VRS, while the green dot in panel (c) indicates the Automatic Weather Station (AWS) at the atmospheric observatory house of VRS. Contour lines in (b) and (c) indicate topographic elevation in m. The blue area in panel (c) outlines the approximate extent of sea ice-free conditions during the field campaign. Fig. 1d-g depict photos of typical different surface cover types during summer. The outlines of FIIC are taken from RGI Consortium (2023). The base layers in panel (a) and (c) are from Landsat satellite imagery provided by Google (2025) and in panel (b) adapted from QGreenland v3.0.0. (Moon et al., 2023). Figures were created in QGIS https://www.qgis.org/de/site/.**

## 2.2 UAV Sounding

Between 27 July and 21 August 2023, 130 vertical temperature profiles up to 500 m above ground level (a.g.l.) were collected during 21 flight days. The profiles were obtained above areas of different surface types, including snow-free tundra, glacier ice, open sea water and a lake (Fig 1c). We used two similar UAVs (DJI Mavic 3 Classic and DJI Mavic Pro), each equipped with iMet-XQ2 sensors, which measure air temperature (accuracy ±0.3 °C), relative humidity (accuracy ±5 %) and position (horizontal accuracy 8 m, vertical accuracy 12 m) at a sampling frequency of 1 Hz. The iMet-XQ2 sensors are widely used for atmospheric soundings with UAVs (InterMet, 2025; Kimball et al., 2020a; Barbieri et al., 2019). To ensure reliable measurements, the positioning of the sensor on the UAV is crucial to avoid heating from propeller downwash or the UAV body and to guarantee sufficient ventilation (Greene et al., 2018; Kimball et al., 2020a). We placed the sensors on the top front of the UAV. Kimball et al. (2020b) demonstrated that this placement is effective, especially if a minimum flight speed of 3 m s$^{-1}$ is maintained to secure sufficient aspiration. After inspection of the soundings, it was found that relative humidity values showed in some cases unrealistically high values after saturation had been reached. To avoid misleading interpretations, the relatively unreliable humidity measurements were excluded from the analysis.

We validated the UAV measurements against air temperature measurements from an atmospheric monitoring mast operated by the Integrated Carbon Observation System (ICOS) at VRS with sensors placed in 20 m and 85 m above ground. To enhance the vertical resolution of the validation data, additional air temperature sensors were installed at 2 m and 8 m above ground on the 16th and 17th of August and in 4.5 m and 0.1 m on the 21st of August 2023 respectively. A total of 65 UAV ascents and descents were conducted adjacent to the mast under varying meteorological. Our results show that about 90 % of the UAV measurements are within a sensor uncertainty range of ±0.6 °C and we identified a slight warming effect during descents, likely caused by propeller downwash and therefore limited our analysis to the data of the ascents.

We further averaged the measurements into 12 m elevation bins to account for the iMet-XQ2 sensor's vertical accuracy, following Hansche et al. (2023). To capture temperature gradients near the surface, we reduced the bin size to 10 m, 5 m, and 3 m for the three lowest altitude ranges. Since the soundings started from the surface, the iMet-XQ2 sensors recorded multiple measurements at the same altitude, increasing vertical resolution and justifying the use of finer bin sizes near the ground.

## 2.3 Comparison with CARRA

The Copernicus Arctic Regional Reanalysis (CARRA) is a state-of-the-art reanalysis product for Arctic regions, based on the convection-resolving and non-hydrostatic weather prediction model HARMONIE-AROME. The model operates at three-hour time intervals with a horizontal resolution of 2.5 km, with data available back to 1991 (Schyberg et al., 2021a). It assimilates an extensive collection of observations around Greenland, including surface observations from Asiaq, the Danish Meteorological Institute (DMI), the Programme for Monitoring of the Greenland Ice Sheet (PROMICE) and the Greenland Climate Network (GC-NET). It also employs 3D variational assimilation of upper-air observations from various sources

such as radiosondes and aircraft (Box et al., 2023). CARRA is laterally forced by ERA5 reanalysis but offers a higher spatial resolution and has several improvements compared to ERA5, including enhanced assimilation of in situ observations and improved incorporation of satellite data, making it particularly suitable for studying meteorological variables at high resolution in the Arctic (Schyberg et al., 2021a; Schmidt et al., 2023). CARRA resolves meteorological variables at 13 height levels up to 500 m a.g.l., including skin temperature and 2 m air temperature. As the extent of the lake is too small to be resolved by CARRA, we selected three fixed grid-point coordinates representing the surface types of glacier ice, open sea water and tundra, which were also sampled with the UAV (Fig. 1). UAV profiles measured above the lake were compared to the adjacent Tundra grid-point. To enable a comparison, both CARRA and UAV profiles were linearly interpolated to a common vertical resolution of 1 m.

To assess the agreement between in situ data and CARRA, we calculated for each profile the mean absolute differences (MAD) and Kendall's tau (τ) rank correlation ideal for non-parametric analysis as an indication of the profile shape similarity. We differentiated between different surface types, vertical layers and the prevailing wind directions, using wind data from an automatic weather station (AWS) approximately 2 km from VRS (Fig. 1).

## 2.4 Drivers of vertical temperature gradients

We first compared the UAV temperature profiles over different surface types to quantify to which extent and height level surface properties influence the atmospheric stratification in our study area. We then made use of the extended time series of CARRA at the fixed grid-point coordinates for its full available period (1991–2024) for the summer months June, July and August (JJA) to assess the drivers of vertical temperature gradients around VRS. We incorporated snow fraction data from CARRA as the key variable modulating tundra surface properties over time (Schyberg et al., 2021b). Van der Schot et al. (2024) demonstrated that CARRA performs well in representing snow cover at VRS, reporting a Pearson correlation coefficient of 0.87 between observed and CARRA-derived snow water equivalent.

We analysed CARRA data for air temperature differences above tundra and glacier ice at daily resolution and correlated them with daily snow fraction anomalies to study the impact of surface properties on the lower troposphere. Next, we applied K-means clustering to categorize the daily vertical temperature gradient profiles derived from the CARRA grid-point above tundra between June and August during the period 1991–2024, interpolated to 1 m vertical resolution. The clustering was conducted for atmospheric layers between the height up to which the troposphere is directly influenced by surface properties and up to 500 m above ground. The lower altitude threshold was set to 100 m a.g.l. based on inspection of the results from correlating snow cover anomalies with anomalies of the temperature difference above tundra and ice (Fig. 3b). Although defining this threshold involves a degree of arbitrariness, a sensitivity test using a lower limit of 200 m (instead of 100 m) yielded remarkably similar synoptic patterns and cluster occurrences (Fig. S2 in the supplement), supporting the robustness of our results with respect to this choice. A lower limit below 100 m was not considered, as altitudes lower than 100 m clearly exhibit direct surface influences (Fig. 3b) and were therefore excluded from the assessment of synoptic-scale

drivers. The resulting clusters were then used to investigate the corresponding large-scale atmospheric conditions using ERA5 reanalysis data (Hersbach et al., 2020).

The K-means clustering method requires the number of clusters (K) to be predefined. Typically, the optimal K is chosen for which adding more clusters results in diminishing reductions of the within cluster distances, while ensuring high inter-cluster separation (Pampuch et al., 2023). Due to the noisiness of climate data, relying solely on the widely used Elbow Method is often insufficient to find a robust optimal number of K (Pampuch et al., 2023). Therefore, we combined the Elbow Method with the Silhouette Score, resulting in five clusters as the optimal choice for our analysis (Rousseeuw, 1987; Wickramasinghe et al., 2022).

### 2.5 Temperature Gradient Clusters and SMB of FIIC

Relating the structure of the lower troposphere to the mass change of FIIC requires SMB data. In the absence of in situ monitoring of FIIC, we use daily output from the Modèle Atmosphérique Régional (MAR) version 3.14. MAR is a regional snow, ice and climate model with a horizontal resolution of 10 km forced at its lateral boundaries by ERA5 (Fettweis, 2007). Its atmospheric component is based on Gallée and Schayes (1994), and the scheme for the surface energy exchanges follows Gallée and Duynkerke (1997). MAR has been extensively validated against in situ data, satellite products and other state-of-the-art regional climate models for Greenland, demonstrating reliable performance for Northeast Greenland, including FIIC (Fettweis et al., 2017; Sutterley et al., 2018; Mattingly et al., 2023).

We analyse the SMB reconstruction together with 2 m air temperatures of FIIC between 1991 and 2024 to assess the sensitivity of mass change to climate warming. The ice cap is segmented into 100 m elevation bins to derive its hypsometry. The equilibrium line altitude (ELA) is estimated annually for each year by linearly interpolating the annual SMB across the defined elevation bands. By projecting the K-means clusters onto the daily SMB from MAR, we isolate how the different temperature gradients relate to the SMB of FIIC.

## 3. Results

### 3.1 UAV profiles and comparison with CARRA

The mean UAV-derived temperature profiles above different surface types (Fig. 2a) reveal that within the lowest 200 m, air temperatures above glacier ice and ice-free ocean surfaces are colder than those above lake and tundra. This contrast peaks near the surface with approximately 3 °C temperature difference. Within the lowest 10 m above ground, the temperature is lowest above glacier ice followed by air above sea water and with the highest temperatures above tundra.

To assess the reliability of CARRA in our study region, we compared its temperature profiles with UAV observations by calculating temperature differences and correlations. The resulting differences, categorized by wind direction and surface type, are shown in fig. 2b. The hourly distribution of UAV sounding times and CARRA timestamps are provided in the supplement (Fig. S3). Across all 130 profiles, the MAD is 1.00 °C and Kendall's tau 0.57, statistically significant ($p < 0.05$)

for all profiles. The largest temperature differences are identified above glacier ice (1.16 °C), while the smallest are observed above the lake (0.77 °C), with no consistent pattern linked to wind direction or surface type. When comparing temperature differences below and above 100 m above ground, the agreement is generally higher above that altitude (MAD = 0.98 °C vs. MAD = 1.12 °C). Kendall's tau is consistently positive across all wind directions and surface types, except for easterly winds, where correlations are negative for all surfaces, averaging -0.20. The temperature differences under easterly winds indicate that CARRA systematically overestimates temperatures within the lowest 100 m by about 1-2 °C, resulting in discrepancies between the UAV and CARRA profiles. When excluding easterly wind directions, the overall correlation coefficient exceeds 0.7.

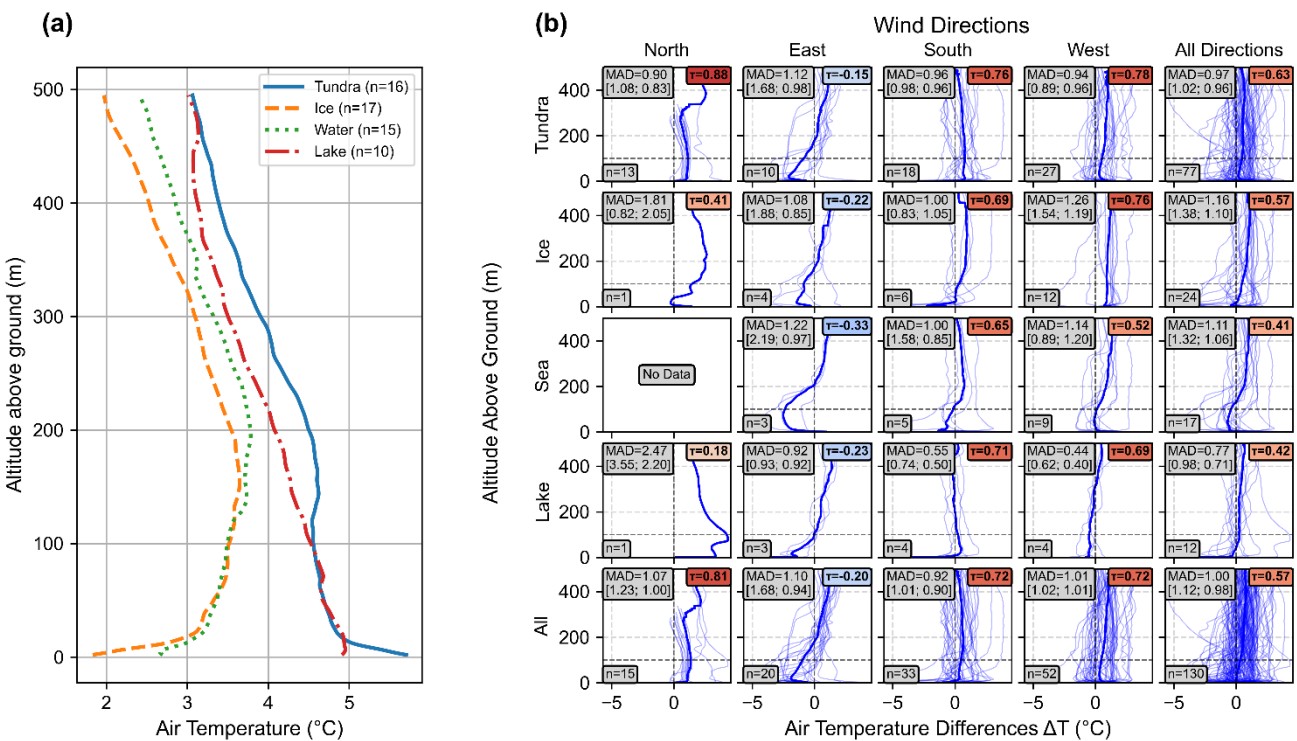

Figure 2: (a) The average of those UAV air temperature profiles reaching from surface up to 495 m a.g.l. over different surface cover types and (b) air temperature differences (ΔT) between UAV and CARRA profiles grouped according to surface cover types (rows) and prevailing wind direction at the AWS (columns). Positive ΔT values indicate higher UAV-measured temperatures compared to CARRA. The thick blue lines show the mean ΔT, the thin lines individual profiles. 10 profiles could not be assigned to a wind direction due to gaps in the AWS data. Annotations show the mean absolute difference (MAD) between 0 m and 500 m, and in square brackets the MAD below and above 100 m, respectively. The number of profiles (n) and Kendall's Tau (τ) are also shown, with τ indicated by colour for clarity.

### 3.2 Drivers of temperature gradients

After validating the reliability of CARRA for our study region, we analyse the full available period from 1991 to 2024 for June, July and August to study the drivers of vertical temperature structures by investigating the influence of surface properties and large-scale synoptic conditions.

### 3.2.1 Surface properties

Analysis of the UAV profiles (Sect. 3.1) indicates that tundra and glacier surfaces exert the strongest influence on the near-surface temperature structure. Therefore, we calculated the daily mean temperature differences of air below 500 m over tundra and glacier ice surfaces, together with the mean snow fraction from 1991 to 2024 (Fig. 3a). Within JJA, the average snow fraction is nearly complete in early June (0.96 on day of year (DOY) 155) and reduces to approximately snow-free conditions by early August (0.1 on DOY 220). Over this period, the temperature differences near the surface increase at the two locations, with higher air temperatures above tundra compared to ice within atmospheric layers up to 100 m a.g.l. The largest temperature differences occur closest to the surface, exceeding 2 °C. As the snow fraction increases again after DOY 220, the air temperature differences above tundra and ice decrease.

To quantify the inverse relationship between snow fraction and temperature difference as an indicator of surface influence on vertical temperature structures, we calculated Pearson correlation coefficients between their daily anomalies (Fig. 3b). Between DOY 155 and DOY 240, we find the strongest statistically significant ($p < 0.01$) negative correlations near the surface, extending up to approximately 100 m above ground. This indicates a strong surface influence on the vertical temperature structure within this layer. Above 100 m a.g.l., the correlation weakens until around 150 m a.g.l., beyond which surface properties appear to have minimal impact on the local vertical temperature profiles.

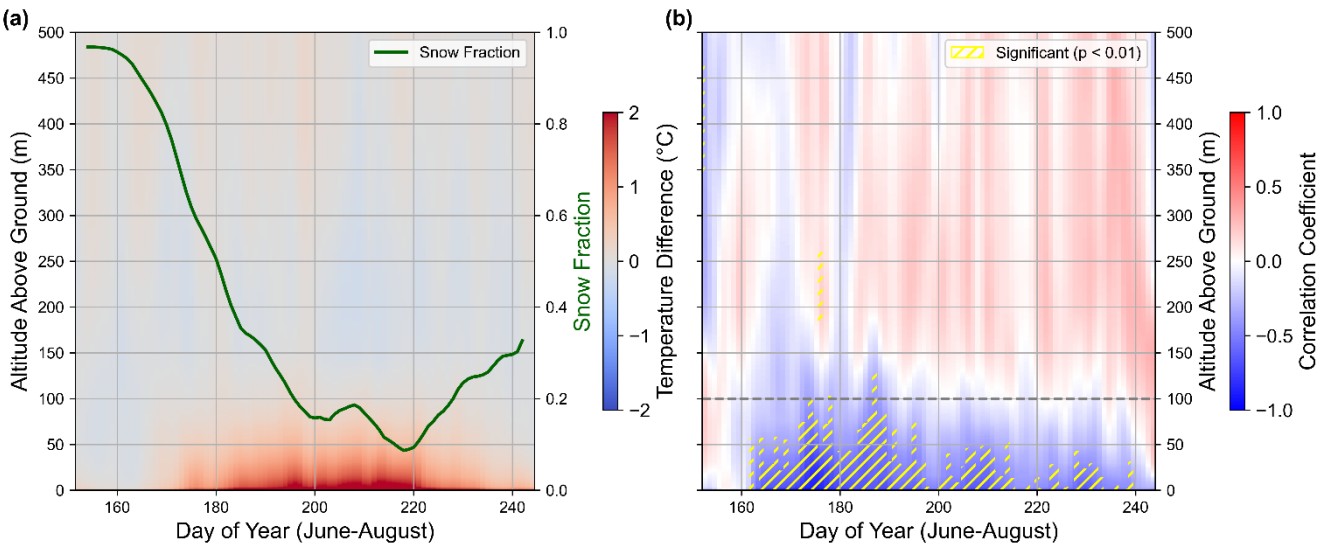

**Figure 3: (a) Mean ΔT of atmosphere profiles a.g,l. between tundra and glacier, smoothed using a Gaussian filter (σ = 1) (colours), and the mean daily snow fraction between 1991 and 2024, smoothed using a 5-day moving average (green line). Positive ΔT indicate higher temperatures above the tundra compared to the glacier. (b) Pearson's correlation coefficient (r) between the daily ΔT anomaly and the daily snow fraction anomaly, smoothed using a Gaussian filter (σ = 1). Statistically significant correlations (p < 0.01) are indicated by hatching. The grey horizontal line at 100 m a.g.l. indicates the height above which K-means clustering was applied. Both figures are only based on CARRA.**

### 3.2.2 Synoptic drivers

To investigate the connection to synoptic drivers, we applied K-means clustering to categorize vertical temperature gradients between 100 m and 500 m a.g.l. into five clusters (Fig. 4). Distinct signatures of atmospheric stratification emerge for each cluster, for the mean temperature gradients (Fig. 4a) as well as for the corresponding mean air temperature profiles (Fig. 4b), with the largest inter-cluster temperature differences occurring around 500 m a.g.l. Apart from distinct vertical profiles, the classification also shows large differences in the number of cluster occurrences, ranging from 1258 for cluster 1 (CL1) to 210 for CL4. CL1 and CL2 are characterized by consistently negative temperature gradients and overall low temperatures. In contrast, CL3 and CL4 and, less pronounced, CL5 show both positive and negative vertical temperature gradients, although with different vertical shapes. CL3 and CL5 are linked to the highest temperatures across all considered elevations, but CL3 is characterized by a negative temperature gradient from 200 to 500 m a.g.l. It should be noted that applying the resulting classification to the temperature difference anomalies between tundra and ice surfaces showed no systematic pattern.

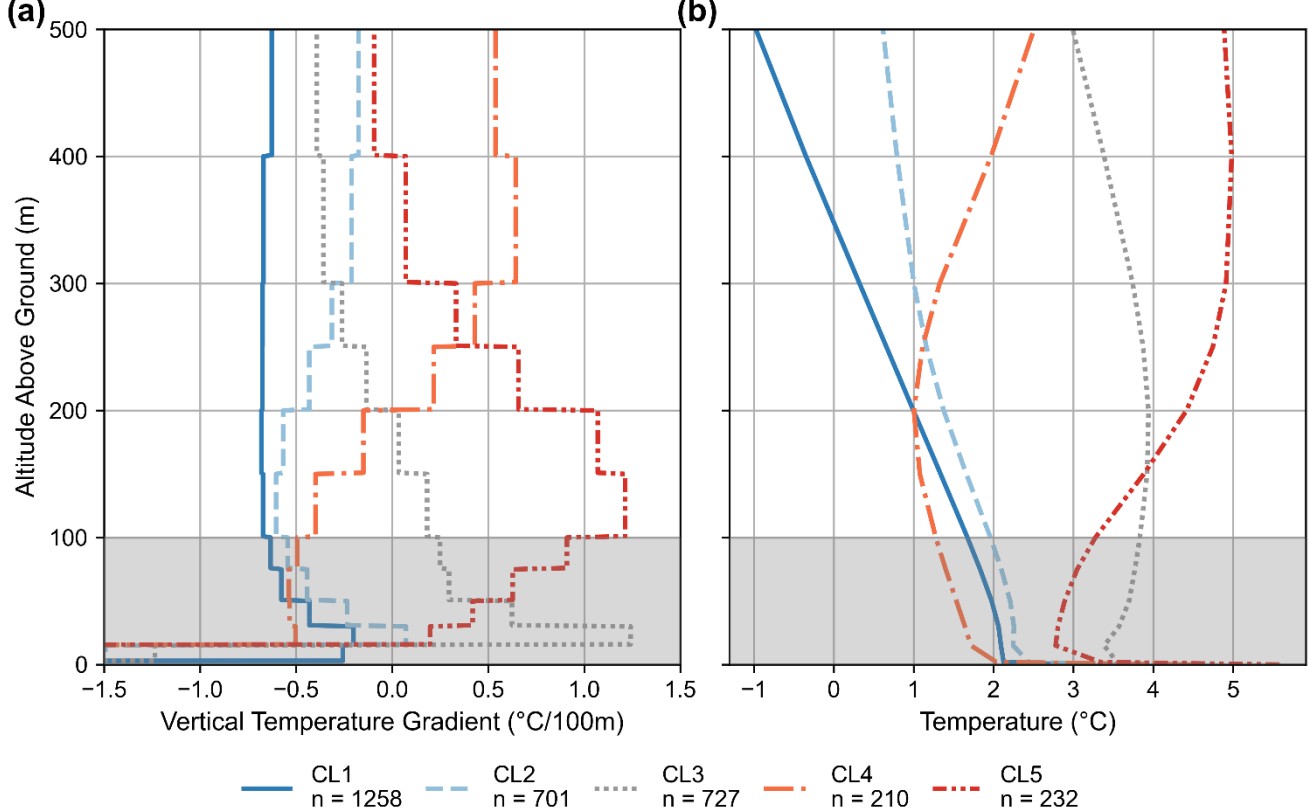

**Figure 4: (a) Vertical temperature gradient and (b) air temperature for each cluster mean in JJA between 1991 and 2024. The shaded grey area indicates the atmospheric layer excluded from clustering due to its strong sensitivity to the properties of the surface cover.**

To link the distinct vertical temperature gradients to synoptic drivers, we calculated composites of large-scale atmospheric conditions for each cluster and used these to assign descriptive names based on the synoptic patterns (Fig. 5a). Different atmospheric conditions emerge, which we in the following can relate to air temperature and wind anomalies. CL1 (Low Pressure), the most frequently occurring cluster during the study period (n=1258), is the only one associated with negative anomalies in both 850 hPa temperature (T_850) and 500 hPa geopotential height (Z_500) within the studied domain (35°W-15°E, 77°N-85°N) and shows northerly wind anomalies in 850 hPa (W_850) at VRS. In contrast, CL2 to CL5 are associated with positive anomalies in Z_500 and T_850, but with varying spatial structures. CL2 (Zonal) and CL4 (Strong Zonal) exhibit the highest positive Z_500 and T_850 anomalies in the northeast of the domain weakening towards the southwest, creating a zonal gradient of pressure anomalies associated with easterly W_850 anomalies at VRS. This gradient is stronger for CL4. CL3 (High Pressure) and CL5 (Strong High Pressure) are characterized by a centre of positive Z_500 anomalies east of VRS, linked to southerly W_850 anomalies at the study site, with CL5 showing stronger anomalies in those variables.

The intraseasonal evolution of cluster occurrence is shown in fig. 5b. CL1 (Low Pressure) with the strongest negative Z_500 and T_850 anomaly appears more frequently in early (DOY ≤ 185) and later summer (DOY ≥ 215) but less often in between. In contrast, CL4 (Strong Zonal) and CL5 (Strong High Pressure), reflecting the strongest positive Z_500 and T_850 anomalies, prevail more frequently in the middle of the summer and CL3 (High Pressure) is more common up until end of July.

Besides intraseasonal variability, we also analysed the long-term temporal evolution of cluster occurrences from 1991 to 2024 (Fig. 5c). All clusters exhibit interannual fluctuations, but CL1 (Low Pressure) shows a notable decline from an occurrence of 45 % during the first half of the time period (1991-2007) to 36 % in the second half (2008-2024). This decrease is offset by minor increases in occurrence of 1-3 % among the other clusters, the strongest in CL5 (Strong High Pressure). We note that during the 21 days of UAV measurements of our field campaign in summer 2023, all clusters except CL4 (Strong Zonal) occurred (Fig. 5d).

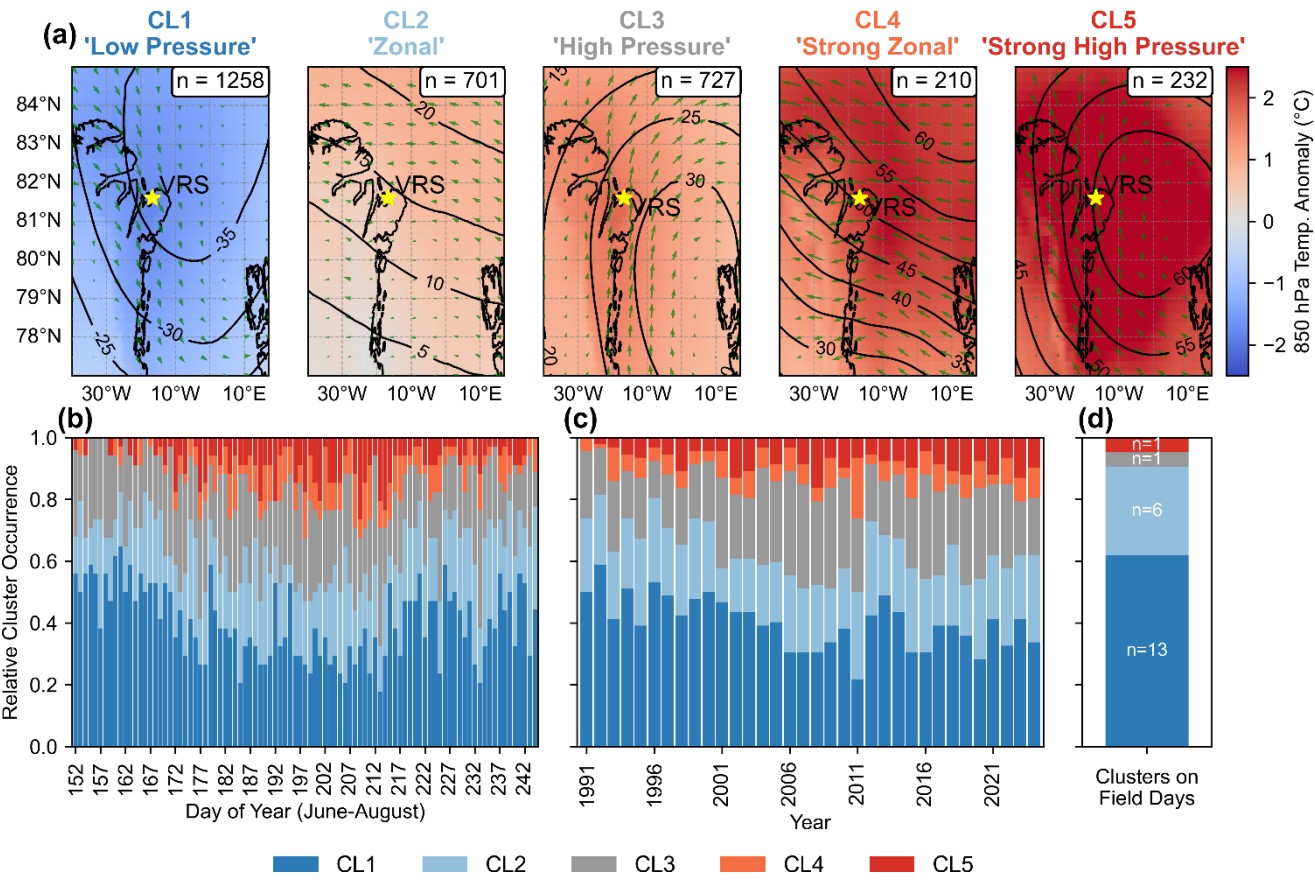

**Figure 5: (a)** Synoptic conditions represented by the cluster means, their descriptive naming and absolute occurrence. Temperature anomaly (T_850) shown by colour, wind anomaly (W_850) by green vectors and 500 hPa geopotential height (Z_500) anomaly in m (contours) from ERA5 reanalysis; **(b)** relative cluster occurrences on the day of year within 1991 to 2024, **(c)** changes in relative cluster occurrence from 1991 to 2024, and **(d)** number of clusters on days of UAV field measurements (Field Days).

### 3.3 SMB of FIIC

We use SMB output from the regional climate model MAR3.14 to study FIIC's SMB over the last decades with respect to atmospheric forcing. Between 1992 and around 2007, the specific annual SMB (SMB per unit area) fluctuates around balance conditions (Fig. 6a), leading to multi-year variations of the cumulative SMB (Fig. 6c). From 2008 onward, annual SMB has become predominantly negative with increasing summer mass losses since 2015, resulting in a cumulative surface mass loss of ~28 Gt from 2008 to 2024.

To investigate the climatic sensitivity of FIIC and to assess hypsometric amplification, we divided the ice cap into 100 m elevation bins. Figure 6b presents the mean annual SMB in each elevation band and for three time periods (1991-2002, 2003-2014, and 2014-2024) along with the ice cap's hypsometry. The ice cap is relatively flat, ranging from sea level to about 1000 m. The latest period 2014-2024 brought a pronounced decline in SMB over all elevations compared to the earlier

periods, associated with an estimated rise of the mean equilibrium line altitude (ELA) from about 376 m to 584 m. Since a large area of FIIC lies within this elevation range, the fraction of positive SMB (accumulation area ratio) declined from 59 % during 1991–2002 to 28 % in 2014–2024. Figure 6d shows the relationship between annual SMB and ELA, with summer (JJA) 2 m temperature anomalies across FIIC. A strong link between summer temperatures and annual SMB is evident, with a Pearson correlation coefficient of -0.88. There is a strong linear relationship between ELAs and annual SMB, however, with stronger decrease in SMB for ELAs around 600 m compared to other ELAs. Furthermore, we find that in 2017, 2019 and 2023 the ELA was around 900 m, indicating that most of the ice cap experienced negative annual SMB.

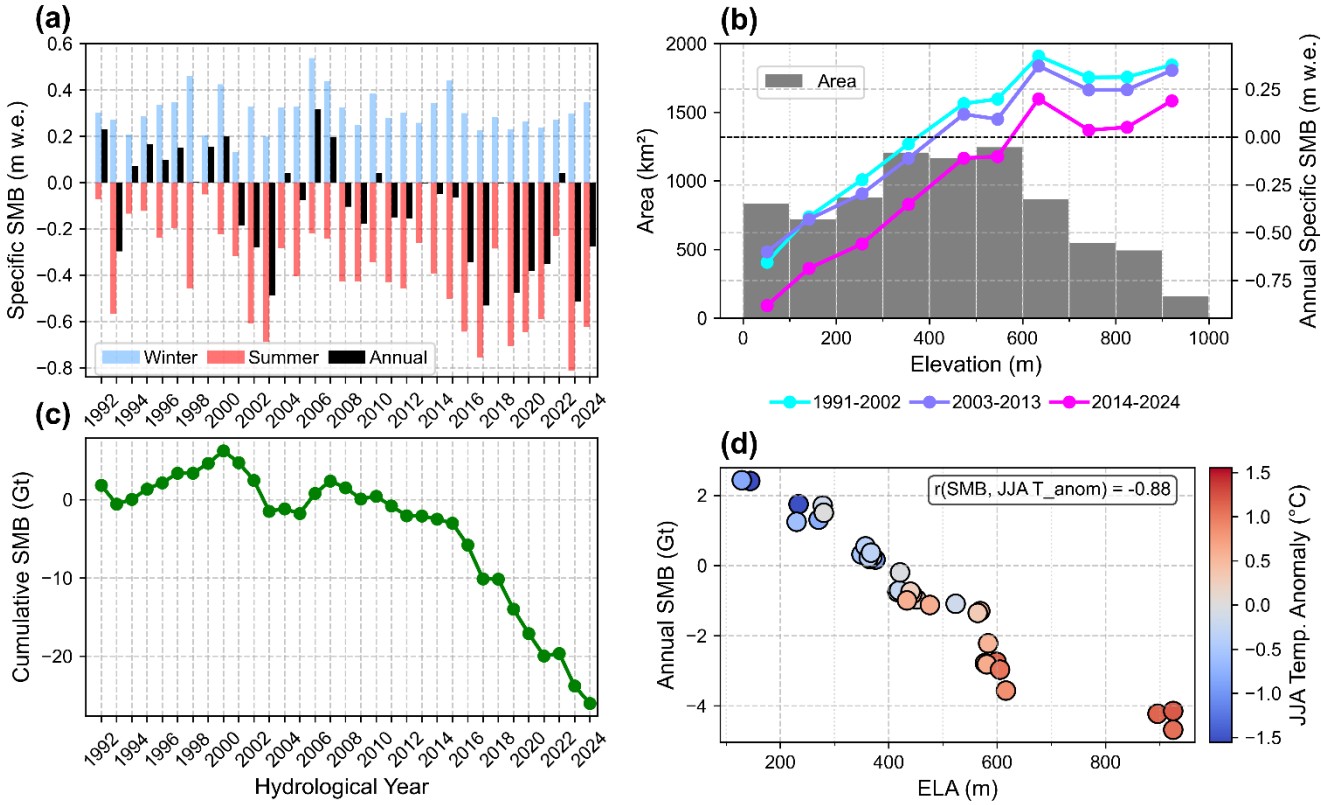

**Figure 6: (a) Specific SMB of FIIC for winter and summer and (c) cumulative SMB over the hydrological years from 1992 to 2024; (b) annual mean SMB for the time periods 1991-2002, 2003-2013 and 2014-2024 and glacier area averaged within each 100 m elevation bin; (d) annual SMB and inferred ELA together with mean 2 m air temperature anomalies (colours) over FIIC during JJA. The Pearson's correlation coefficient (r) of annual SMB and 2 m air temperatures is annotated in the diagram. The three dots marking the highest ELA and lowest SMB refer to the years 2017, 2019 and 2023.**

### 3.4 Temperature gradients and SMB of FIIC

To study the relationship between vertical temperature gradients and SMB, we computed the daily SMB of FIIC for each atmospheric cluster. Both the temporal evolution of mean SMB during summer (Fig. 7a) and the mean summer SMB within the 100 m elevation bins (Fig. 7b) show distinct impacts related to the atmospheric clusters. CL1 (Low Pressure) is

consistently associated with higher mean SMB values throughout summer and across all elevations. In contrast, CL5 (Strong High Pressure) shows the lowest SMB values, also across all elevations. CL2 (Zonal), CL3 (High Pressure) and CL4 (Strong Zonal) exhibit intermediate SMB values. CL5 (Strong High Pressure), representing the highest T_850 anomalies, accounts for 16.0 % of the total summer mass loss while occurring relatively rarely (7.4 %). Similarly, the relative contributions to the summer mass loss of CL3 (High Pressure) and CL4 (Strong Zonal) are higher than their ratio of occurrence. CL1 (Low Pressure) occurs most frequently (40.2 %) and contributes 17.4 % to the total summer mass loss, while the contribution of CL2 (Zonal, 20.9 %) is like its ratio of occurrence (22.4 %).

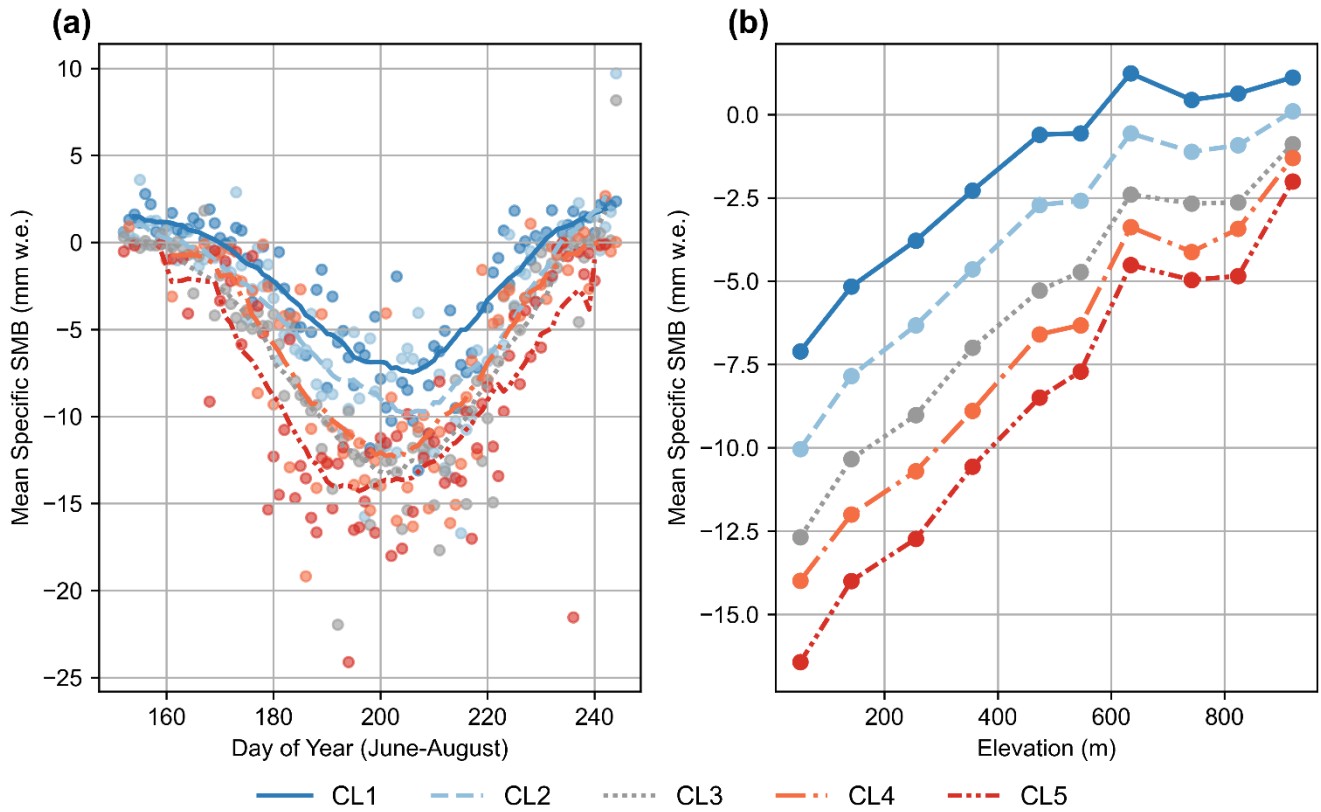

**Figure 7: (a) Mean summer SMB of FIIC over the day of year from 1991 to 2024 for each cluster (circles) and with the values smoothed by a 25-day moving mean (lines). (b) Daily mean summer SMB for 1991 to 2024 aggregated into 100 m elevation bins and separated for each cluster.**

The total SMB of FIIC is strongly related to summer air temperature anomalies (Fig. 6d). To better understand the drivers of its increased mass loss, we analysed the annual mean summer SMB (Fig. 8a) and air temperatures (Fig. 8b) for each cluster in the study period and estimated linear trends. All clusters except CL1 (Low Pressure) show significant ($p < 0.05$) negative SMB trends in summer, ranging from approximately -5.0 to -7.5 mm day$^{-1}$ over the study period (Fig. 8a). These trends correspond to significant positive trends ($p < 0.05$) in summer air temperatures for CL2 through CL5 which increase by about 1.0 °C to 1.2 °C from 1991 to 2024 (Fig. 8b). In contrast, CL1 (Low Pressure) does not exhibit a

significant linear trend in either variable. The strong relationship between air temperature and SMB is further supported by significant Pearson correlation coefficients of –0.91 for all clusters (p < 0.05). This relationship also holds true within each cluster, as reflected by correlation coefficients for each subset of data ranging from -0.47 to -0.86 (Fig. 8c).

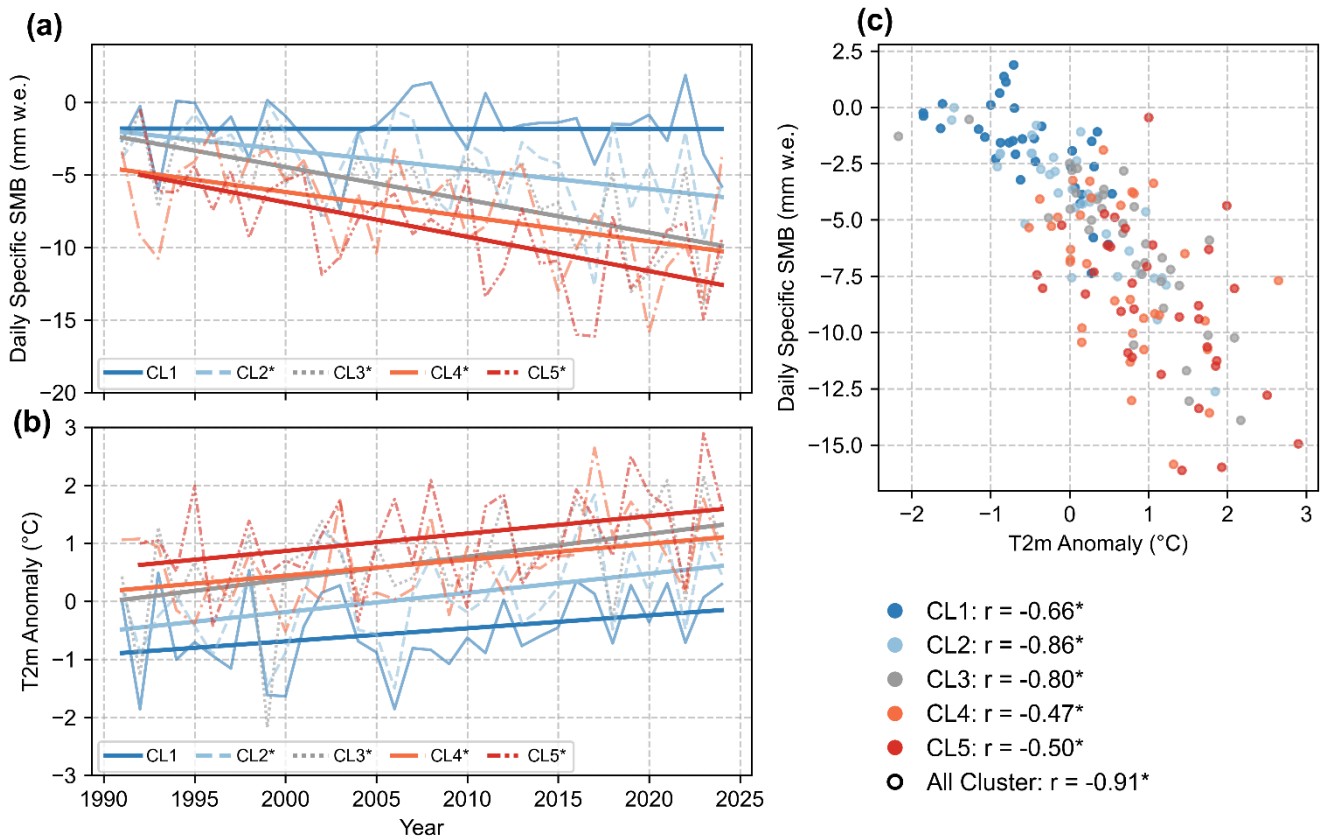

**Figure 8: (a) Daily average summer SMB of FIIC and (b) corresponding 2 m air temperature (T2m) anomalies, shown per cluster**
**from 1991 to 2024. Linear trends are indicated by straight lines, with statistically significant trends (p < 0.05) marked by asterisks in the legend; (c) daily average summer SMB and 2 m air temperature (T2m) anomaly per cluster. Pearson's correlation coefficients (r) are included in the legend with asterisks marking statistical significance (p < 0.05).**

## 4. Discussion

### 4.1 UAV and CARRA profiles

Our in situ data demonstrate clear influence of surface properties and associated surface warming on near-surface air temperatures, particularly within the lowest 100 m of the troposphere. Air masses above ice and open water are consistently colder, whereas those above tundra surfaces exhibit higher near-surface air temperatures (Fig. 2a). The profiles further indicate that above a certain altitude, the lapse rate is consistent across all clusters, though the threshold altitude above which this typical lapse rate is present depends on surface type (throughout the profile above the lake, and above approximately 200

320 m for the other surfaces). However, since the profiles in fig. 2a comprise varying numbers of soundings collected at different times and meteorological conditions, this composite offers limited insights about underlying processes. We therefore refrain from drawing further conclusions regarding atmospheric boundary-layer processes from these data. Nevertheless, the observed pattern of surface influence on air temperatures near the surface aligns with expected surface energy exchange processes over different surface types (Oerlemans and Grisogono, 2002; Shahi et al., 2023). Furthermore, the profile

characteristics above ice cannot be generalized to the whole ice cap, as the soundings were obtained from close to its margin. Higher elevations are topographically less sheltered, favouring stronger synoptic scale influences on the lower troposphere, and different energy exchange processes may prevail there (Oerlemans et al., 1999).

When comparing UAV soundings with CARRA, it is important to note that AWS data from VRS are assimilated in CARRA (Copernicus Climate Change Service, ECMWF). Consequently, while CARRA 2 m air temperature can be

expected to match well, comparisons of the vertical structure up to 500 m above ground give independent information on the quality of the reanalysis product. We found a MAD of 1.00 °C across all profiles with 0.14 °C higher agreement above 100 m compared to below indicating that representing the influence of surface properties on air temperature variability may be challenging for CARRA at our study site. Further, the agreement between UAV and CARRA temperature profiles between 0 m and 500 m varies with meteorological conditions and surface type. The largest temperature differences are observed above

the ice surface, while the smallest are found above the lake. For all wind directions, we note significantly ($p < 0.05$) positive correlation values of Kendall's Tau except for easterly wind conditions, under which CARRA systematically overestimates air temperatures within the lowest 100 m. For both easterly and northerly winds, air masses are advected over glacier ice or sea ice before reaching the study sites (Fig. 1c), where they can cool due to energy exchanges with the frozen surface. This site-specific meteorological effect of horizontal movement of cooled air masses may not be adequately resolved by CARRA

at the local scale, which could explain the observed discrepancies to the UAV measurements. Furthermore, katabatic winds descending from the ice cap may interact with larger-scale flows and could influence the observed discrepancies, highlighting subjects for future investigations. Excluding profiles measured during easterly wind conditions, the agreement between UAV and CARRA profiles improves to a Kendall's Tau exceeding 0.7, which is comparable to findings of Hansche et al. (2023) in southeast Greenland.

We acknowledge that the comparison of CARRA with our UAV soundings is restricted to the range of meteorological conditions sampled; soundings could not be conducted during precipitation or at wind speeds exceeding 12 m s[-1], constraining our data collection to 21 days. The sampling range is further limited by the time of day, as most UAV profiles were obtained between 09:00 and 16:00 local time, with no observations during nighttime (23:00-08:00; Fig. S3 in the supplement). In addition, low clouds that may have affected air temperature measurements were not documented; however,

cloud cover variability at the local and temporal scales of our soundings cannot be fully captured by CARRA. Nevertheless, cloud occurrence and characteristics play an important role for local surface energy exchanges and near-surface air temperatures and should be further investigated with dedicated approaches.

Despite these limitations, our comparison suggests that CARRA reproduces the vertical atmospheric temperature structure around VRS between 0 m and 500 m above ground reasonably well for the sampled meteorological conditions (MAD = 1.00°C, $\tau$ = 0.57). Considering the remoteness of the region and scarcity of in situ meteorological data in Northeast Greenland, CARRA provides a valuable estimate of local air temperature conditions and appears suitable for studying vertical temperature gradients over its reanalysis period.

### 4.2 Drivers of vertical temperature gradients

In both CARRA and UAV data, we are able to show an important influence of surface conditions on air temperature differences over tundra and glaciated surfaces below 100 m above ground. Furthermore, we have shown that over the melt period in summer, air temperature differences between glacier and tundra covary with the extent of the snow cover (Fig. 3). This indicates that snow-free tundra surfaces absorb more solar radiation, warming the near-surface air, while glaciated surfaces remain at or below 0 °C due to surface melt (Oerlemans and Grisogono, 2002). This contrast can be particularly pronounced during clear-sky conditions in summer, when shortwave radiative input enhances surface heating (Ignatiuk et al., 2022). The observed relationship between snow fraction anomalies and temperature differences below 100 m above ground is a consequence of the snow-albedo feedback. This feedback is a key mechanism in amplifying Arctic warming (Thackeray et al., 2019) and is hereby locally quantified in terms of its vertical extent in the atmospheric column. Our findings underscore the critical role of snow cover by modulating the surface energy balance. While shorter periods of snow cover have been observed in the Arctic during spring and summer, no significant trends have yet been reported around VRS (Derksen and Mudryk, 2023; Mohammadzadeh Khani et al., 2022; van der Schot et al., 2024). This may be due to the region's cold climate, where warming could increase snowfall, offsetting enhanced melt (Thackeray et al., 2019). Nevertheless, warming in the study area along with its proximity to the Arctic Ocean, which exhibits retreating sea ice extent and increasing precipitation, makes it highly sensitive to environmental changes, including changing snow cover duration (Bennett et al., 2024; Day et al., 2013; Box et al., 2019).

To investigate temperature structures decoupled from surface influence, we applied K-means clustering to categorize vertical temperature gradients between 100 m and 500 m a.g.l. Atmospheric composites of T_850 and Z_500 for each cluster reveal that temperature structures above 100 m are tightly connected to large-scale circulation patterns (Fig. 5a). The synoptic regime of CL1 is characterized by the lowest T_850 anomalies corresponding to northerly W_850 anomalies, resulting in the lowest air temperatures and the most negative temperature gradients (< -0.5 °C 100 m$^{-1}$) between 100 m and 500 m a.g.l. at VRS (Fig. 4, Fig. 5a). In contrast, CL4 and CL5 exhibit temperature inversions in their mean vertical profiles, which appear linked to the highest T_850 values and to the advection of relatively warm air masses by easterly and southerly winds. CL2 and CL3 show intermediate vertical profiles corresponding to the weakest pressure and wind anomalies in their respective synoptic regimes. These results demonstrate that above 100 m a.g.l., distinct vertical temperature structures emerge under different large-scale circulation patterns. We conclude that while the snow-albedo feedback dominates near-

surface temperatures below 100 m above ground, large-scale synoptic conditions control vertical atmospheric structures at higher levels. This agrees with an estimate of a very shallow boundary layer at VRS (Gryning et al., 2023).

Analysis of the cluster occurrences (Fig. 5b–d) shows that most in situ soundings were conducted during low-pressure conditions (CL1), with only two field days falling into high-pressure regimes (CL3, CL5). This constrains the ability of the in situ profiles to capture the variability of vertical temperature structures during summer. However, validating CARRA using in situ soundings and utilising an extended CARRA time period provides a valuable way to bypass this limitation and to derive climatological insights beyond the field campaign.

### 4.3 SMB of FIIC

FIIC has experienced a predominantly negative SMB since 2008, with sharply increased mass loss after 2015, driven by intensified summer ablation (Fig. 6). We examined the relationship between SMB, air temperature, the ice cap's hypsometry, and its link to the atmospheric clusters. In North Greenland warming exceeded 1.2 °C since 1991 (Khan et al., 2022; Rantanen et al., 2022). We found a strong and significant correlation ($r = -0.88$, $p < 0.05$) between annual SMB of the ice cap and summer temperature anomalies, confirming warming temperature as a driver of surface mass loss (Fig. 6d). However, the accelerated mass loss reflects a high climatic sensitivity of the ice cap, likely linked to its hypsometry. The ice cap is relatively flat, and our analysis shows that warming has caused the ELA to rise to higher elevations characterized by the largest glacier area. Therefore, moderate additional warming causes disproportionate increase in surface melt. Similar patterns have been observed for other flat glaciers and icefields (Davies et al., 2024). When the ELA rises above these broad low-slope elevations, the albedo declines and a melt-elevation feedback can be triggered, both acting as positive feedbacks that amplify surface mass loss (Davies et al., 2024; McGrath et al., 2017). FIIC's high climatic sensitivity due to its hypsometry is particularly pronounced as a disproportionate drop in SMB around ELAs near 600 m above sea level (Fig. 6d). Furthermore, Noël et al. (2017) found a sharp reduction in the capacity of the firn layer to buffer runoff by refreezing meltwater after 1997 across Greenland's peripheral glaciers and ice caps, especially in North Greenland. This reduced refreezing capacity has increased runoff and could contribute to the recent, intensified mass loss at FIIC as well.

We found the highest annual mass loss of FIIC in the years 2017, 2019, and 2023 corresponding to ELAs located above most of the ice cap (Fig. 6). The Greenland Ice Sheet also experienced above-average mass loss in 2019 and 2023. However, surface mass loss in 2017 was only modest (Mankoff et al., 2021; Poinar et al., 2023). The Ice Sheet's SMB is largely driven by anomalies in its southwest sector and FIIC is located northeast of the Ice Sheet, which may explain the discrepancies in their SMB signals (Briner et al., 2020). Glaciers and ice caps in Svalbard have experienced negative mass balance anomalies in 2017 and 2019 as well, suggesting similar large-scale atmospheric drivers for the SMB in both regions (Sasgen et al., 2022).

## 4.4 Temperature gradients and SMB of FIIC

The temperature gradient clusters show distinct impacts on the SMB of FIIC, each linked to different synoptic conditions (Fig. 5, 7). The synoptic patterns align with their expected effects on SMB. CL1 (Low Pressure), associated with the lowest air temperatures, corresponds to a trough-like pattern enabling northerly air advection around FIIC, and results in the highest SMB values. In contrast, CL5 (Strong High Pressure) is linked to the highest air temperatures and strong positive $Z\_500$ anomalies centred east of FIIC, resulting in southerly $W\_850$ anomalies that provide warm air advection causing the strongest melting (Fig. 4, Fig. 5, Fig. 7).

Despite larger interannual variability, shifts in the frequency of atmospheric clusters emerge over time. The occurrence of CL1 (Low Pressure) decreased by 9 % in the second half of the study period, while CL5 (Strong High Pressure) showed the largest increase. Given the strong link between the clusters inferred locally and synoptic conditions, these shifts are likely part of changes in large-scale atmospheric circulation over Greenland. Increased westerly flow over North Greenland has been associated with reduced cloud cover and foehn-like conditions on the lee side of the Greenland Ice Sheet expanding to FIIC (Sasgen et al., 2022; Noël et al., 2019; Mattingly et al., 2023). The clusters associated with melt-favouring synoptic conditions and enhanced surface mass loss (CL3, CL4, and CL5) occurred more frequently than average during the years of highest mass loss (2017, 2019, 2023) at FIIC (Fig. 5c). This suggests an additional dynamic atmospheric contribution to the recent increase in surface mass loss of the ice cap.

Long-term trends in SMB and air temperatures differ notably when grouping the time series by cluster (Fig. 8). For CL1 (Low Pressure), neither SMB nor air temperatures show significant temporal trends between 1991 and 2024 ($p < 0.05$). In contrast, the time series associated with the other clusters consistently show declining SMB and warming. Based on the analysed data sets alone it is difficult to explain why CL1 does not also show a significant trend in the impacts. However, the trough-like structure of $Z\_500$ anomalies and low $T\_850$ associated with CL1 reflect northerly air advection from the Arctic Ocean, where perennial summer sea ice extent has remained relatively stable, compared to more pronounced retreat in the Greenland and Barents Seas (Bennett et al., 2024; Müller et al., 2022; Onarheim et al., 2018). As a result, air masses from the north could be subject to more stable conditions of energy exchange with the ocean surface, while those advected by southerly and easterly winds encounter increasingly more open water as sea ice retreats, leading to enhanced moisture uptake and possibly warmer and more melt-inducing conditions.

Given the ongoing Arctic warming and FIIC's SMB response, its ELA will further rise in coming decades exposing large areas for sustained melt. Continued sea ice retreat and warming ocean waters could enhance the energy exchange with the ocean surface and accelerate ice shelf disintegration, reducing buttressing forces and thereby leading to increased dynamic mass loss from outlet glaciers (Möller et al., 2022; Bendtsen et al., 2017). These processes are likely to drive substantial long-term reduction of the ice cap's volume. Additionally, the melt-elevation feedback may impede future regrowth, potentially pushing FIIC's mass loss towards a tipping point (Davies et al., 2024).

## 5. Conclusion

We found overall robust agreement of CARRA's representation of vertical temperature profiles and UAV measurements around VRS. Systematic differences were observed when the measured air masses had been advected over proximal ice surfaces. Both datasets consistently show the modulation of near-surface air temperatures with surface cover types of tundra, glacier and open water. Using daily CARRA data (1991-2024), we demonstrate that the snow-albedo feedback particularly controls summer temperature structures in the atmospheric column up to 100 m above ground. Beyond this level, vertical temperature profiles are primarily shaped by large-scale atmospheric circulation.

Using regional climate model output, we showed that FIIC has experienced increased surface mass loss since 2015, driven by rising summer air temperatures. From 1991 to 2024, the ELA has risen by around 200 m, exposing an increasingly large area of the ice cap to net ablation, amplified by a flat hypsometry. This process may reduce the albedo over an increasingly large area, enhancing surface melt and contributing to FIIC's high climatic sensitivity.

We were able to classify five distinct vertical temperature structures by K-means clustering and relate those to large-scale circulation to show their respective impact on the SMB. Atmospheric conditions linked to particularly high surface melt have become more frequent in recent decades and have favoured record mass loss and higher ELA's. We argue that these shifts are connected to observed changes in the large-scale circulation over North Greenland. Future warming will continue to raise the ELA of the ice cap and trigger a melt-elevation feedback that could drive long-term deglaciation of the region. Further research on atmospheric feedbacks involving snow cover, ice cover and sea ice retreat is important for understanding the interactions with the terrestrial cryosphere, as well as future glacier mass loss and their role in Arctic amplification.

### Data availability

The UAV soundings are publicly available at https://zenodo.org/records/15690233?preview=1&token=eyJhbGciOiJIUzUxMiJ9.eyJpZCI6ImQ1NGYyNzBiLTA1YTEtN DI0YS05YTI4LWM2Y2Q3NDkyM2M4MSIsImRhdGEiOnt9LCJyYW5kb20iOiI5MjVmNjE5NDAwZjYyMzQ3MTI3ODDJ jZTExNDY1Yjk5YyJ9.gjyhj_PAxi4F0jZrJ8Dp9WIgs5guQh5w215R2HBWBwL2eQBI64nLxUzf56l4JxSf0DJ7F4tQvKRM _dx31FsXaA. CARRA and ERA5 data are provided by the European Centre for Medium-Range Weather Forecasts, available at https://cds.climate.copernicus.eu/. Output from MAR version 3.14. are available upon request from Xavier Fettweis (https://www.uliege.be/cms/c_9054334/en/directory?uid=u204898). The AWS data from the atmospheric observatory house can be downloaded from https://www2.dmu.dk/asiaqmet/Default.aspx. Data from the ICOS atmospheric monitoring mast can be obtained from the ICOS data portal at https://data.icos-cp.eu/portal/. The Python code for data download and for producing the figures is available upon request from the corresponding author.

**Author contributions**

JF and WS conducted the fieldwork at VRS. The study was designed and carried out by JF with input from JA and IS. JF performed the data analyses. JA and IS discussed, shaped und supervised the study progress. JF prepared the first draft of the manuscript, and all co-authors discussed and contributed to the final manuscript.

**Competing interests**

The corresponding author declares that none of the authors has any competing interests.

**Acknowledgements**

We gratefully acknowledge funding from the EU Horizon 2020 project INTERACT for the Transnational Access funding of the fieldwork (Grant agreement No 871120; LowTropVRS). IS acknowledges funding by the Helmholtz Climate Initiative REKLIM (Regional Climate Change), a joint research project of the Helmholtz Association of German Research Centres (HGF). JA acknowledges funding from the Austrian Science Fund (FWF) project P35388. We are especially thankful to Xavier Fettweis for providing the output of the Modèle Atmosphérique Régional (MAR). We sincerely thank Asiaq – Greenland Survey and the Integrated Carbon Observation System (ICOS) for their valuable atmospheric monitoring efforts at VRS. We are also grateful to Aarhus University and the VRS team for enabling and supporting the fieldwork. We thank the European Centre for Medium-Range Weather Forecasts (ECMWF) for providing access to ERA5 and CARRA datasets. Financial support for article publication was provided by the University of Graz. This work was supported by the Helmholtz Association under Program-Oriented Funding (PoF-IV), within the research program "Changing Earth – Sustaining our Future", Topic 2 "Ocean and Cryosphere in Climate".

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
