# Peer review of "The vertical structure of the troposphere and its connection to the surface mass balance of Flade Isblink in Northeast Greenland"

_EGUsphere, 2025_

## Author Comment (AC2)

**Final author comments**

Dear Editor, Dear Referees,

We sincerely thank you for the two valuable reviews and the helpful editorial guidance. We are pleased by the interest in our work and greatly appreciate your time and effort. We are confident that by incorporating your comments, we can compile a much more mature manuscript.

However, while we appreciate many valuable comments, we felt in parts offended by RC1. For example, the first Reviewer notes that this is the first study by an early career scientist and links that to the judgement of immaturity (poor writing, riddled with typos, lack in storytelling, ...) of the work. We welcome criticism of the manuscript, including criticism that labels it as poorly written or immature. Putting this in the context of the main author's career stage is inappropriate and should not play a role in scientific discourse, as it is the science quality that we should address and not personal situations. Additionally, we had the impression that our scientific storyline had been partly misunderstood in RC1 and we in some occasions missed necessary clarification of where in the manuscript certain impressions arose (for example see below the first specific comment referring to the framing of the article addressing the Greenland ice sheet).

In the following, the referees' comments are given in **bold**, and our responses are provided in *italic*. All amendments that we will make in a revised manuscript are *highlighted in green*. We trust that our revisions and clarifications will satisfactorily address the concerns raised, and we hope to be invited to submit a revised manuscript.

Once again, we thank you much for the valuable input!

All the best,

Jonathan Fipper, on behalf of the author team

**Review Comment 1 (RC1)**

To the editor and authors of "The vertical structure of the troposphere and its connection to the surface mass balance of Flade Isblink in Northeast Greenland"

The authors in this manuscript report the results of 130 soundings by UAV at the Villum Research Station located in Northern Greenland. Their objective is to connect model realizations of surface mass balance to vertical atmospheric profiles of temperature as well as to evaluate the ability of a reanalysis product to represent those temperatures across several "surface types". In the current state of this manuscript, I believe the authors are unable to accomplish this objective cleanly and I am rejecting the manuscript. Additionally, the writing in

this manuscript is poor, carries with it a lack in storytelling, riddled with typos, and isn't yet to the quality of scientific publication. I recognize that this is the first manuscript of an early career scientist so I wish to get across that they shouldn't be entirely discouraged. There is a good paper in this work that is worth writing. I encourage the first author to bring this work back to the drawing board and the end product will be something they will be proud of. The below list is non-exhaustive and does not include any technical corrections, but I have compiled some of the red flags which stuck out to me about the manuscript:

The introduction of the text leads me towards an expectation that this manuscript is going to comment specifically on glacier ice sheet mass balance. I agree with the authors on the importance of understanding surface mass balance (SMB) in this context. They likewise mention SMB in other parts of the manuscript, characterizing the ice loss in larger regions of the ice sheet. Why then is 3/4th of the analysis on non-glaciated parts of the VRS? The framing of the manuscript needs to be rewritten, with emphasis not on a major uncertainty of the Greenlandic Ice Sheet, but on the background needed for what the authors scope of work can comment on.

Our work links large-scale synoptic patterns, which emerge from the analysis of local conditions at the Villum Research Station (VRS), with the surface mass balance (SMB) of the Flade Isblink Ice Cap (FIIC) – a local ice cap. Therefore, local conditions in non-glaciated areas are not directly linked to the SMB of the Greenland ice sheet.

RC1 refers to "larger regions of the ice sheet" and "major uncertainties of the Greenland Ice Sheet" when showing the framing of the article. We agree that addressing the ice sheet's SMB is clearly beyond the scope of this work. However, we are relating local atmospheric conditions to the SMB of the peripheral ice cap Flade Isblink and not the ice sheet. Having read the introduction, we are wondering how the impression of focusing on the ice sheet arose or whether a misunderstanding is present here? However, in a revised version we will clarify the focus towards the non-glaciated areas and a peripheral ice cap even stronger in order to make the aim of the work clear.

One major pillar of this manuscript if the comparison of UAV temperature profiles is to the CARRA reanalysis product. CARRA assimilates weather station data as part of its reanalysis product. The authors fail to report that VRS is itself a weather station included in the data assimilation

(Figure 2.2.7.1,

https://confluence.ecmwf.int/display/CKB/Copernicus+Arctic+Regional+Reanalysis+(CARRA): +Full+system+documentation ). Now data assimilations are never one-to-one matches, but there is significantly less value in evaluating the utility of a reanalysis at the location of its tie points. If a reanalysis is accurate, it is most accurate at the location of a tie point. Still, VRS is only a ground station tie point and thus the vertical profile might still be worth addressing. Regardless, this needs to be acknowledged or used as a guide to direct additional analysis.

Thank you for raising this excellent point. Demonstrating the reliability of CARRA using near-surface observations at VRS is not our focus. It is unique to have the meteorological monitoring mast at VRS and we do not only compare CARRA 2 m conditions with ground observations but relate CARRA's height levels to vertical profiles from UAV soundings. We will take this into account by adding a short section in the discussion (section 4.1).

The authors have a large operational filter on their dataset that is acknowledged but then disregarded. They are limited by precipitation and wind speed. They may also be limited by time of day (i.e. waking hours) but that isn't reported. They likewise do not distinguish by clouds, either low clouds which may disrupt their sampling or by higher clouds, which like low clouds would profoundly affect the surface energy balance at the time of sounding. Despite this limitation, the authors claim on Line 311 that "CARRA represents the vertical atmospheric structure around VRS well". This is entirely inaccurate and not sufficiently reductive to the evidence the authors have to make such a statement. A correct statement might be something such as "Our observation-reanalysis comparison shows that CARRA is accurate in temperature (MAD = XXXX) within XXXX-XXXX m AGL during 00:00 – 00:00 on clear sky days." Anything less reductive cannot be demonstrated with the supplied analysis.

We agree that cloudiness characteristics play an important role in surface energy exchanges and their influence on temperature profiles. However, cloud conditions are highly variable in both space and time and can in certain cases not be represented in CARRA at the local and temporal scales of our soundings adequately. We will address this by explicitly reflecting on the limitations of CARRA in representing such locally differing conditions in the discussion section 4.1.. To avoid introducing misleading or poorly constrained interpretations, we will not include cloudiness in our statistical assessment. We will restrict our statements about the observation-reanalysis agreement to the times during which we sounded. Given that one focus of the study is the role of surface types and their influence on vertical temperature structures, we will further differentiate our assessment by surface type and wind direction. As shown in a later comment, we define a limit up to which temperature structures are significantly influenced by surface properties and we will use this limit to further refine our evaluation of the observation—reanalysis comparison by distinguishing between those vertical layers.

The authors utilize an iMet-XQ2 sensor onboard a multirotor UAV to profile the atmosphere. Having used the same combination myself, I know this is an apt choice. That said, what is missing such that I am baffled it isn't included in the analysis, is the humidity measurement that comes along with the temperature measurement. Humidity is as key of an atmospheric state variable as temperature and likewise just as important to understanding the energy balance of the surface and near-atmosphere such that it is inappropriate to be excluded from the analysis.

Thanks again, this is an important concern. We investigated the humidity measurements and found that, when they reached approximately 100%, consecutive measurements appeared to be inaccurate, showing unreasonably high values (not reducing plausibly to levels below saturation again despite reaching drier air masses). To avoid misleading interpretations, we have decided not to include these relatively unreliable measurements in our analysis and will add an explanation in the method section 2.1 in a revised manuscript.

The analysis does not include a metrological discussion and interpretation of atmospheric soundings. Discussion on, for example, the location of a surface layer, is missing from the analysis. This omission shines through when the authors arbitrary choose 100m as the lower limit of data excluded from clustering. How was that altitude determined? In Figure 2 (a) the authors show selected average profiles (related: averaged and selected how exactly?) that do not relax to the typical lapse rate until about 200 m for plotted profiles. Why use 100m? The remedy for this is a careful appraisal of atmospheric structure that is part of the reported text.

Discussing a surface layer explicitly is a valuable suggestion that we will consider in a revised manuscript. We will reflect in more detail on the atmospheric structures shown by the averaged profiles in fig. 2a. Regarding the averaging and selection of these profiles, we will clarify that the averaging was performed only for profiles that extend up to 495 m above ground level, as not all soundings reached this height. Due to the resulting samples covering differing meteorological conditions, the interpretation should be made carefully as to what these profiles can tell on the climatological scale. This is also the reason, why we opt for careful evaluation of CARRA with in situ measurements and then obtaining climatological information from the reanalysis dataset.

The decision for the upper altitude limit of surface impact (100 m) should indeed be well founded. We stress, that our ambition in this part is to apply an objective k-means clustering in order to display synoptic conditions, regardless of the surface conditions — which is the main motivation for finding this upper limit. In our study, we choose this altitude (100 m) by informed inspection of figure 3a and 3b, which, as RC1 notes, involves a degree of arbitrariness. We will make this explicit in a revised version in the methods section. We considered in depth, whether we should apply some statistical method to derive this elevation, however, any statistical derivation of this limit would also require arbitrary choices (E.g., which p-value is chosen for significance testing? How is the upper limit of surface control for the full summer period defined?). We therefore rather suggest to add considerations on the sensitivity of our choice in the revised version: To test the robustness of our upper limit of 100 m with respect to its influence on the k-means clustering results, we now additionally conducted k-means clustering for an upper limit of 200 m (instead of 100 m). The resulting cluster occurrences and associated synoptic patterns are shown in fig. R1 below in this response both for the original choice of 100 m and the one for 200 m. While some differences are expected the overall synoptic structures remain remarkably consistent, which we

take as a support of little influence of which exact elevation we use as the lower boundary for the clustering method. An upper limit lower than 100 m should not be considered, since fig. 3b shows that altitudes below 100 m include very clear surface influences – information that should be excluded from clustering when inferring the synoptic drivers of vertical structures.

Figure R1: Synoptic conditions and temporal characteristics of cluster occurrence derived from K-means clustering of daily CARRA temperature gradient profiles above tundra between 1991 and 2024. The left panels show results for excluding the lowest 100 m (upper limit of surface control) from clustering, while the right panels show the corresponding results for an upper limit of surface control of 200 m. In both cases: (a) synoptic conditions represented by the cluster means, their descriptive naming, and absolute occurrence, with temperature anomaly (colour) and 500 hPa geopotential height anomaly (m) from ERA5 reanalysis; (b) relative cluster occurrences by day of year (1991–2024); (c) changes in relative cluster occurrence between 1991 and 2024; and (d) number of clusters corresponding to days of UAV field measurements ("Field Days").

The authors utilize the MAR model to produce the SMB for VRS. This is an appropriate choice. That said, MAR also includes atmospheric temperature. Why not also include a MAR comparison? The reason for doing so is clear. The authors note that wind direction-based anomalies in CARRA are due to upwind surface impacts and that such effects "may not be resolved in CARRA". First, I'd like to note that the "may" here can definitely be resolved, as CARRA has extensive documentation. I encourage the authors to spend more time investigating why CARRA could have such a mismatch at a more technical level. Regardless, if the authors chose to use MAR for the quality of its surface mass balance, then it is also the tool to test the impact of easterly winds on vertical profiles.

We considered comparing MAR temperature data with UAV soundings. However, as stated in the manuscript in L107 and L144, the MAR version 3.14 we utilized has a much lower spatial resolution than CARRA (horizontally 10 km vs 2.5 km). Therefore, the observed regional pattern during easterly winds is on a subgrid scale in MAR, limiting its ability to resolve these processes and making it less suitable than using CARRA for that task.

Further, we would like to challenge the term "upwind" that is used in the comment as we suspect a misunderstanding here. While we hypothesize that eastern wind directions advect air masses cooled down by surface energy exchanges above ice (L304-306) and hence refer to horizontal movement of cooled air masses, we suspect that the RC refers to "upwind" as a vertical/convective process. In order to avoid confusion we will in a revised version clarify the horizontal movement of the process that we discuss by rewriting L305-L306 as: For both easterly and northerly winds, air masses are advected over glacier ice or sea ice before reaching the study sites (Fig. 1c), where they can cool due to energy exchanges with the frozen surface. This site-specific meteorological effect of horizontal movement of cooled air masses may on that spatial scale not be resolved by CARRA, which could explain the observed discrepancies to the UAV.

The authors use K-means clustering to group atmospheric observations. This is an accepted use. However, they fail to mention with what? K-means clustering is typically used in situations which highly multivariable data, though the authors only present data for temperature. My guess is that they do so on CARRA data, as they later define clusters by regional pressures. Either way, this needs to be explicitly discussed in the text before clusters can be evaluated.

To our assessment, we do not fail to mention with which variable we performed the clustering: We state in L131-L133 "[...] we applied K-means clustering to categorize temperature gradients of the vertical profiles [...]". However, we acknowledge that the K-means clustering method can be presented more clearly and we will expand on that by rewriting the section L131-L134 as follows: Next, we applied K-means clustering to categorize the daily vertical temperature gradient profiles derived from CARRA above tundra between June and August for the period 1991–2024, interpolated to 1 m vertical resolution. The Clustering was conducted for atmospheric layers above the height up to which the troposphere is directly influenced by surface properties, up to 500 m. The resulting clusters were then used to investigate the corresponding large-scale atmospheric conditions using ERA5 reanalysis data (Hersbach et al., 2020).

The authors concluded that surface albedo affects the surface mass balance at their sampling site. Albedo isn't a new result and I would hope that the authors would develop a more quantitative description of albedo at their site. Also, given the weight of the importance placed on the concept of snow-albedo feedback, I would expect the word "albedo" to show up earlier in the background or methods rather than for the first time in the discussion on Line 335.

It is true that albedo affects surface mass balance, and as the comment states, this is not a new result; however, we feel a bit misunderstood here, as we are not sure where the Reviewer's impression arises that we state "that surface albedo affects the surface mass balance at their sampling site". What we indeed discuss in section 4.3 are effects amplifying melt due to the flat hypsometry of FIIC and refer to Davies et al. (2024) when considering ice cap albedo as a relevant feedback mechanism in such a system (L346-L350). The snow-albedo feedback is not directly

related to surface mass balance in our analysis. We wonder as to where in the manuscript this impression arose and would appreciate a clarification.

It is a good advice to introduce "albedo" earlier in the study and we will do so already in the introduction in a revised version.

The authors have a tendency to overuse verbiage with value judgements included. Example such as Line 29 "unequivocally" or Line 61 "exceptionally" are unnecessary and not appropriately reductive to the presented and supported science in the manuscript.

We will revise our verbiage and will improve the precision when using adjectives. However, we believe that the use of "unequivocally" in the context of scientific consensus (L29) is relevant and appropriate, as it clarifies that there is a scientific consensus that anthropogenic greenhouse gas emissions are the primary cause of climate warming. This term is also commonly used in high-level scientific compilations to make similar statements (e.g. IPCC, 2021; Siegert et al., 2025). We will, however, critically revise qualitative wording and omit this wherever possible in order to maximize objectiveness.

The authors tend to report relationships without quantifying them. Examples such as Line 62: What height is near surface, when is the coldest month. Line 112: What improvements? Line 129: How well?

Thanks for pointing this out. We will follow the advice and include quantifications at the given examples and wherever else appropriate.

**Review Comment 2 (RC2)**

**Summary**

In this study, Fipper and coauthors analyze the vertical temperature structure above Flade Isblink Ice Cap (FIIC) in northeast Greenland during summer and its influence on the ice cap's surface mass balance (SMB). They first use a large number of atmospheric profiles measured by uncrewed aerial vehicles (UAVs) over difference surface types near Villum Research Station to relate the near-surface thermal gradients to surface type and assess the accuracy of the Copernicus Arctic Regional Reanalysis (CARRA) in reproducing these temperature structures. Finding a good agreement between CARRA and the UAV profiles, they use CARRA to extend the analysis in time and assess the synoptic circulation patterns that control the vertical temperature structure over FIIC. They find that the decreasing SMB trend for FIIC is related to summer atmospheric warming across most circulation patterns, and that the anticyclonic patterns that favor increased surface melt have shown an increasing trend over time.

In my assessment, this is an excellent study overall. The paper is well-written and easy to follow, with clear figures that illustrate the main findings. The use of UAV data to validate the CARRA reanalysis is innovative and the methods for this comparison appear robust. I have some comments below asking for some minor clarifications and technical corrections. Once these comments are addressed, I feel this paper will be a valuable addition to the literature on the coupling of the atmosphere and cryosphere in Greenland.

Thanks a lot for the kind words and the overall positive assessment. This gives us courage and makes us proud.

**Minor comments**

- General comment: Although no UAV measurements were taken at the top of the FIIC, it might be helpful to have some discussion of how applicable these results are to the ice cap in this topographically complex region. Do the authors expect that the vertical temperature profiles and gradients over the higher elevations of the ice cap would be similar to those observed at lower elevations?

Thank you, this is an interesting point! We do not expect the vertical temperature profiles and temperature gradients at higher elevations to be similar to those at our sites, because steeper slopes compared to the sounding locations facilitate katabatic winds and in general, the near-surface climate is likely very different in higher elevations, less topographically protected environments and different surface energy exchange processes may prevail. However, we do not study this in detail and a comprehensive analysis of vertical profiles across FIIC is out of the scope of our study, which is based on local in situ profiles at VRS. However, we will discuss this by referring to literature in a revised manuscript.

And in using the CARRA data for the long-term analysis (e.g. L125–127), are the CARRA data only from the three fixed grid points, or are data from a larger spatial domain around FIIC examined?

The data for the long-term analysis are from the three fixed grid points (Fig. 1). We will point this out more clearly in a revised manuscript.

- L24–25: There appears to be a discrepancy between the abstract and the findings presented in Section 3.4. The abstract states that FIIC mass loss is driven by "rising summer air temperatures for all synoptic conditions", while L286–287 states that cluster 1 (low pressure) does not exhibit a significant linear trend in either summer mean SMB or summer air temperature.

This is an excellent point that indeed requires clarification. With the statement in the abstract, we want to point out that summer conditions are driving annual surface mass balance. While we report a Pearson correlation coefficient of -0.88 (p

- ~L215: It would be nice to have some more discussion about what causes the varying vertical temperature gradients and temperature profiles across clusters, as shown in Fig. 4. Unless I missed it, there isn't really any physical explanation provided for why these distinct shapes of the temperature profile occur for each cluster

Those temperature gradients and temperature profiles emerge from clustering the gradients between 100 m and 500 m above tundra surface based on CARRA data. We show that at those altitudes synoptic conditions drive vertical temperature structures and we assign descriptive names for the clusters' synoptic conditions (Fig. 5). But indeed, we do not specifically link synoptic conditions with the vertical temperature structures and will take this up in a revised manuscript. We will examine the synoptic wind directions and wind speeds by adding wind vectors to fig. 5a (also see comment below). Since wind plays a key role in the mixing of air masses and the formation of temperature inversions, this addition will provide a more comprehensive understanding of the processes shaping the vertical temperature structures shown in fig. 4.

- ~L237 (Fig. 5): I suggest the authors consider plotting wind vectors on the map panels of Fig. 5 (Fig. 5a), to more directly show the synoptic flow direction. For example, L364 discusses northerly wind advection during the cluster 1 regime that could be shown more directly with composite wind vectors.

We do agree that would be a valuable addition and we will do that in a revision.

- L245: How is SMB per unit area different from summing SMB across all grid cells on the ice cap?

The SMB per unit area results in the specific mass balance, expressing the average SMB over the ice cap over a time interval (e.g. in meters water equivalent). Summing the SMB across all grid cells (including a mass conversion) results in a total mass change over a time interval (e.g. in Gt/yr).

- L304–305: Could there also be a downsloping effect from FIIC that influences the disagreement between CARRA and UAV profiles for the easterly wind regime?

That is a very interesting question that we considered as well. In our assessment, downslope flows from FIIC may indeed interact with synoptic easterly winds, potentially influencing the vertical structures observed at our sites and contributing to the disagreement with CARRA. This process would, however, be consistent with our hypothesis of the advection of air masses cooled by surface energy exchanges over frozen surfaces. We will add this point to the discussion as it highlights avenues for future research.

- L309–311: Also, it appears that the the overwhelming majority of the field days were categorized into Cluster 1 (low pressure) or Cluster 2 (zonal) synoptic flow, with little sampling of the anticyclonic conditions? (see Fig. 5d)

This is correct and a valid constraint that any campaign of a limited time period will have. We will discuss this constraint in greater detail and the only way we can think of in order to bypass it is to stress the reasonable agreement with CARRA and then rely on CARRA for longer-term perspectives as we do in this study.

**Technical corrections**

- L250: This reference to Fig. 6c should instead be to Fig. 6b?

We agree and will adapt that.

- L273: "rare" --> "rarely"

We agree and will adapt that.

- L349–350: Something about the grammar is off in this sentence... should this be written instead as "...pronounced as \*a\* disproportional drop..."?

We agree and will adapt that.

- L380: "Why not also CL1 shows" --> "why CL1 does not also show"

We agree and will adapt that.

**References**

- Davies, B., McNabb, R., Bendle, J., Carrivick, J., Ely, J., Holt, T., Markle, B., McNeil, C., Nicholson, L., and Pelto, M.: Accelerating glacier volume loss on Juneau Icefield driven by hypsometry and melt-accelerating feedbacks, Nat Commun, 15, 5099, https://doi.org/10.1038/s41467-024-49269-y, 2024.
- IPCC: Framing, Context, and Methods, in: Climate Change 2021 The Physical Science Basis: Contribution of Working Group I to the Sixth Assessment Report of the Intergovernmental Panel on Climate Change, edited by: IPCC, Cambridge University Press, 147–286, https://doi.org/10.1017/9781009157896.003, 2021.
- Siegert, M., Sevestre, H., Bentley, M. J., Brigham-Grette, J., Burgess, H., Buzzard, S., Cavitte, M., Chown, S. L., Colleoni, F., DeConto, R. M., Fricker, H. A., Gasson, E., Grant, S. M., Gulisano, A. M., Hancock, S., Hendry, K. R., Henley, S. F., Hock, R., Hughes, K. A., Karentz, D., Kirkham, J. D., Kulessa, B., Larter, R. D., Mackintosh, A., Masson-Delmotte, V., McCormack, F. S., Millman, H., Mottram, R., Moon, T. A., Naish, T., Nath, C., Orlove, B., Pearson, P., Rogelj, J., Rumble, J., Seabrook, S., Silvano, A., Sommerkorn, M., Stearns, L. A., Stokes, C. R., Stroeve, J., and Truffer, M.: Safequarding the polar regions from dangerous geoengineering: a critical assessment of proposed concepts and future prospects, Front Sci, 3, https://doi.org/10.3389/fsci.2025.1527393, 2025.
- van der Schot, J., Abermann, J., Silva, T., Rasmussen, K., Winkler, M., Langley, K., and Schöner, W.: Seasonal snow cover indicators in coastal Greenland from in situ observations, a climate model, and reanalysis, The Cryosphere, 18, 5803–5823, https://doi.org/10.5194/tc-18-5803-2024, 2024.

---

## Author Response (AR1)

Dear Editor and Referees,

We thank the reviewers for their detailed and thoughtful comments, the editorial advice and appreciate the valuable time put into this. We have carefully considered all suggestions and implemented most of them in the revised manuscript. We are convinced that these revisions have substantially improved the clarity and scientific robustness of our work and led to a more mature manuscript.

In the following, the referees' comments are given in **black and bold**, our responses are provided in *black and italic* and extracts from the manuscript in green. If not differently specified, line numbers refer to the revised manuscript. In the revised version, we provide supplement material, to which is also referred to in this response and in the revised manuscript.

Thank you for considering our re-submission!

All the best,

Jonathan Fipper, on behalf of the author team

**Point by point reply to review comment 1 (RC1)**

**To the editor and authors of "The vertical structure of the troposphere and its connection to the surface mass balance of Flade Isblink in Northeast Greenland"**

**The authors in this manuscript report the results of 130 soundings by UAV at the Villum Research Station located in Northern Greenland. Their objective is to connect model realizations of surface mass balance to vertical atmospheric profiles of temperature as well as to evaluate the ability of a reanalysis product to represent those temperatures across several "surface types". In the current state of this manuscript, I believe the authors are unable to accomplish this objective cleanly and I am rejecting the manuscript. Additionally, the writing in this manuscript is poor, carries with it a lack in storytelling, riddled with typos, and isn't yet to the quality of scientific publication. I recognize that this is the first manuscript of an early career scientist so I wish to get across that they shouldn't be entirely discouraged. There is a good paper in this work that is worth writing. I encourage the first author to bring this work back to the drawing board and the end product will be something they will be proud of. The below list is non-exhaustive and does not include any technical corrections, but I have compiled some of the red flags which stuck out to me about the manuscript:**

**The introduction of the text leads me towards an expectation that this manuscript is going to comment specifically on glacier ice sheet mass balance. I agree with the authors on the importance of understanding surface mass balance (SMB) in this context. They likewise mention SMB in other parts of the manuscript, characterizing the ice loss in larger regions of the ice sheet. Why then is 3/4${}^{th}$ of the analysis on non-glaciated parts of the VRS? The framing of the manuscript needs to be rewritten, with emphasis not on a major uncertainty of the Greenlandic Ice Sheet, but on the background needed for what the authors scope of work can comment on.**

*Our work links large-scale synoptic patterns, which emerge from the analysis of local conditions at the Villum Research Station (VRS), with the surface mass balance (SMB) of the Flade Isblink Ice Cap (FIIC) – a local ice cap and not the Greenland Ice Sheet.*

*RC1 refers to "larger regions of the ice sheet" and "major uncertainties of the Greenland Ice Sheet" when showing the framing of the article. We agree that addressing the ice sheet's SMB is clearly beyond the scope of this work. Having read the introduction, we are wondering how the impression of focusing on the ice sheet arose or whether a misunderstanding is present here? However, in the revised version we clarify the focus towards the non-glaciated areas and a peripheral ice cap stronger as follows in L50-L53:*

This study combines UAV-based soundings of the lower troposphere at Villum Research Station (VRS), taken over different surface types (sea water, glacier, tundra, lake), with regional climate

model output and reanalysis data to investigate vertical temperature structures and their connection to synoptic circulation patterns, as well as the surface mass balance (SMB) of the peripheral Flade Isblink Ice Cap (FIIC).

**One major pillar of this manuscript if the comparison of UAV temperature profiles is to the CARRA reanalysis product. CARRA assimilates weather station data as part of its reanalysis product. The authors fail to report that VRS is itself a weather station included in the data assimilation (Figure 2.2.7.1, https://confluence.ecmwf.int/display/CKB/Copernicus+Arctic+Regional+Reanalysis+(CARRA): +Full+system+documentation ). Now data assimilations are never one-to-one matches, but there is significantly less value in evaluating the utility of a reanalysis at the location of its tie points. If a reanalysis is accurate, it is most accurate at the location of a tie point. Still, VRS is only a ground station tie point and thus the vertical profile might still be worth addressing. Regardless, this needs to be acknowledged or used as a guide to direct additional analysis.**

*Thank you for raising this excellent point. Demonstrating the reliability of CARRA using near-surface observations at VRS is not our focus. It is unique to have the meteorological monitoring mast at VRS and we do not only compare CARRA 2 m conditions with ground observations but relate CARRA height levels to vertical profiles from UAV soundings. We took this into account by adding the following to section 4.1 in L328-L331:*

When comparing UAV soundings with CARRA, it is important to note that AWS data from VRS are assimilated in CARRA (Copernicus Climate Change Service, ECMWF). Consequently, while CARRA 2 m air temperature can be expected to match well, comparisons of the vertical structure up to 500 m above ground give independent information on the quality of the reanalysis product.

**The authors have a large operational filter on their dataset that is acknowledged but then disregarded. They are limited by precipitation and wind speed. They may also be limited by time of day (i.e. waking hours) but that isn't reported. They likewise do not distinguish by clouds, either low clouds which may disrupt their sampling or by higher clouds, which like low clouds would profoundly affect the surface energy balance at the time of sounding. Despite this limitation, the authors claim on Line 311 that "CARRA represents the vertical atmospheric structure around VRS well". This is entirely inaccurate and not sufficiently reductive to the evidence the authors have to make such a statement. A correct statement might be something such as "Our observation-reanalysis comparison shows that CARRA is accurate in temperature (MAD = XXXX) within XXXX-XXXX m AGL during 00:00 – 00:00 on clear sky days." Anything less reductive cannot be demonstrated with the supplied analysis.**

*We agree that cloudiness characteristics play an important role in surface energy exchanges and their influence on temperature profiles. However, cloud conditions are highly variable in both*

*space and time and can in certain cases not be represented in CARRA at the local and temporal scales of our soundings. We address this by explicitly reflecting on the limitations of CARRA in representing such locally differing conditions in the discussion section 4.1 in L349-L352:*

In addition, low clouds that may have affected air temperature measurements were not documented; however, cloud cover variability at the local and temporal scales of our soundings cannot be fully captured by CARRA. Nevertheless, cloud occurrence and characteristics play an important role for local surface energy exchanges and near-surface air temperatures and should be further investigated with dedicated approaches.

*To avoid introducing misleading or poorly constrained interpretations, we do not include cloudiness in our statistical assessment. We restrict our statements about the observation-reanalysis agreement to the times during which we sounded and provide the distribution of points in time at which soundings were collected in fig. S3 of the supplement. Given that one focus of the study is the role of surface types and their influence on vertical temperature structures, we further differentiate our assessment by surface type and wind direction. As shown in a later comment, we define a limit up to which temperature structures are significantly influenced by surface properties and we use this limit to further refine our evaluation of the observation–reanalysis comparison by distinguishing between those vertical layers.*

*We have rewritten the section 4.1 and address all these points in L331-L357:*

We found a MAD of 1.00 °C across all profiles with 0.14 °C higher agreement above 100 m compared to below indicating that representing the influence of surface properties on air temperature variability may be challenging for CARRA at our study site. Further, the agreement between UAV and CARRA temperature profiles between 0 m and 500 m varies with meteorological conditions and surface type. The largest temperature differences are observed above the ice surface, while the smallest are found above the lake. For all wind directions, we note significantly ($p < 0.05$) positive correlation values of Kendall's Tau except for easterly wind conditions, under which CARRA systematically overestimates air temperatures within the lowest 100 m. For both easterly and northerly winds, air masses are advected over glacier ice or sea ice before reaching the study sites (Fig. 1c), where they can cool due to energy exchanges with the frozen surface. This site-specific meteorological effect of horizontal movement of cooled air masses may not be adequately resolved by CARRA at the local scale, which could explain the observed discrepancies to the UAV measurements. Furthermore, katabatic winds descending from the ice cap may interact with larger-scale flows and could influence the observed discrepancies, highlighting subjects for future investigations. Excluding profiles measured during easterly wind conditions, the agreement between UAV and CARRA profiles improves to a Kendall's Tau exceeding 0.7, which is comparable to findings of Hansche et al. (2023) in southeast Greenland.

We acknowledge that the comparison of CARRA with our UAV soundings is restricted to the range of meteorological conditions sampled; soundings could not be conducted during precipitation or at wind speeds exceeding 12 m s$^{-1}$, constraining our data collection to 21 days. The sampling range is further limited by the time of day, as most UAV profiles were obtained between 09:00 and 16:00 local time, with no observations during nighttime (23:00-08:00; Fig. S3 in the supplement). In addition, low clouds that may have affected air temperature measurements were not documented; however, cloud cover variability at the local and temporal scales of our soundings cannot be fully captured by CARRA. Nevertheless, cloud occurrence and characteristics play an important role for local surface energy exchanges and near-surface air temperatures and should be further investigated with dedicated approaches.

Despite these limitations, our comparison suggests that CARRA reproduces the vertical atmospheric temperature structure around VRS between 0 m and 500 m above ground reasonably well for the sampled meteorological conditions (MAD = 1.00°C, τ = 0.57). Considering the remoteness of the region and scarcity of in situ meteorological data in Northeast Greenland, CARRA provides a valuable estimate of local air temperature conditions and appears suitable for studying vertical temperature gradients over its reanalysis period.

**The authors utilize an iMet-XQ2 sensor onboard a multirotor UAV to profile the atmosphere. Having used the same combination myself, I know this is an apt choice. That said, what is missing such that I am baffled it isn't included in the analysis, is the humidity measurement that comes along with the temperature measurement. Humidity is as key of an atmospheric state variable as temperature and likewise just as important to understanding the energy balance of the surface and near-atmosphere such that it is inappropriate to be excluded from the analysis.**

*Thanks again, this is an important concern. We investigated the humidity measurements and found that, when they reached approximately 100%, consecutive measurements appeared to be inaccurate, showing unreasonably high values (not reducing plausibly to levels below saturation again despite reaching drier air masses). To avoid misleading interpretations, we have decided not to include these relatively unreliable measurements in our analysis. We added an explanation in the method section 2.1 L91-L93:*

After inspection of the soundings, it was found that relative humidity values showed in some cases unrealistically high values after saturation had been reached. To avoid misleading interpretations, the relatively unreliable humidity measurements were excluded from the analysis.

**The analysis does not include a metrological discussion and interpretation of atmospheric soundings. Discussion on, for example, the location of a surface layer, is missing from the**

**analysis. This omission shines through when the authors arbitrary choose 100m as the lower limit of data excluded from clustering. How was that altitude determined? In Figure 2 (a) the authors show selected average profiles (related: averaged and selected how exactly?) that do not relax to the typical lapse rate until about 200 m for plotted profiles. Why use 100m? The remedy for this is a careful appraisal of atmospheric structure that is part of the reported text.**

*Discussing a surface layer explicitly is a valuable suggestion that we considered thoroughly when revising our manuscript. In the revised version we reflect in more detail on the atmospheric structures shown by the averaged profiles in fig. 2a. Regarding the averaging and selection of these profiles, we clarify in the figure caption (L185-L186), that the averaging was performed only for profiles that extend up to 495 m above ground level, as not all soundings reached this height. Due to the resulting samples covering differing meteorological conditions, the interpretation should be made carefully as to what these profiles can tell on the climatological scale. This is also the reason, why we opt for careful evaluation of CARRA with in situ measurements and then obtaining climatological information from the reanalysis dataset.*

*We address this now more clearly in L317-L324:*

The profiles further indicate that above a certain altitude, the lapse rate is consistent across all clusters, though the threshold altitude above which this typical lapse rate is present depends on surface type (throughout the profile above the lake, and above approximately 200 m for the other surfaces). However, since the profiles in fig. 2a comprise varying numbers of soundings collected at different times and meteorological conditions, this composite offers limited insights about underlying processes. We therefore refrain from drawing further conclusions regarding atmospheric boundary-layer processes from these data. Nevertheless, the observed pattern of surface influence on air temperatures near the surface aligns with expected surface energy exchange processes over different surface types (Oerlemans and Grisogono, 2002; Shahi et al., 2023).

*The decision for the upper altitude limit of surface impact (100 m) should indeed be well founded. We stress, that our ambition in this part is to apply an objective method to link vertical air temperature structures to synoptic conditions, regardless of the surface conditions – which is the main motivation for finding this upper limit. In our study, we choose this altitude (100 m) by informed inspection of figure 3a and 3b, which, as RC1 notes, involves a degree of arbitrariness. We considered in depth, whether we should apply some statistical method to derive this elevation, however, any statistical derivation of this limit would also require arbitrary choices (e.g., which p-value is chosen for significance testing? What measure of variation is chosen to derive the upper limit of surface influence for the full summer period?). We therefore rather add considerations on the sensitivity of our choice in the revised version. To test the robustness of our upper limit of 100 m with respect to its influence on the k-means clustering results, we now additionally conducted*

*k-means clustering for an upper limit of 200 m (instead of 100 m). The resulting cluster occurrences and associated synoptic patterns are shown in fig. S2 in the supplement both for the original choice of 100 m and the one for 200 m. While some differences are expected, the overall synoptic structures remain remarkably consistent, which we take as a support of little influence of which exact elevation we use as the lower boundary for the clustering method. A limit lower than 100 m should not be considered, since fig. 3b shows that altitudes below 100 m include clear surface influences – information that should be excluded from clustering when inferring the synoptic drivers of vertical structures.*

*We address this explicitly in the methods section of the revised manuscript in L137-L145:*

The clustering was conducted for atmospheric layers between the height up to which the troposphere is directly influenced by surface properties and up to 500 m above ground. The lower altitude threshold was set to 100 m a.g.l. based on inspection of the results from correlating snow cover anomalies with anomalies of the temperature difference above tundra and ice (Fig. 3b). Although defining this threshold involves a degree of arbitrariness, a sensitivity test using a lower limit of 200 m (instead of 100 m) yielded remarkably similar synoptic patterns and cluster occurrences (Fig. S2 in the supplement), supporting the robustness of our results with respect to this choice. A lower limit below 100 m was not considered, as altitudes lower than 100 m clearly exhibit direct surface influences (Fig. 3b) and were therefore excluded from the assessment of synoptic-scale drivers.

**The authors utilize the MAR model to produce the SMB for VRS. This is an appropriate choice. That said, MAR also includes atmospheric temperature. Why not also include a MAR comparison? The reason for doing so is clear. The authors note that wind direction-based anomalies in CARRA are due to upwind surface impacts and that such effects "may not be resolved in CARRA". First, I'd like to note that the "may" here can definitely be resolved, as CARRA has extensive documentation. I encourage the authors to spend more time investigating why CARRA could have such a mismatch at a more technical level. Regardless, if the authors chose to use MAR for the quality of its surface mass balance, then it is also the tool to test the impact of easterly winds on vertical profiles.**

*We considered comparing MAR temperature data with UAV soundings. However, as stated in the preprint manuscript in L107 and L144, the MAR version 3.14 we utilized has a much lower spatial resolution than CARRA (horizontally 10 km vs 2.5 km). Therefore, the observed regional pattern during easterly winds is on a subgrid scale in MAR, limiting its ability to resolve these processes and making it less suitable than using CARRA for that task.*

*Further, we would like to clarify the term "upwind" that is used in the comment. While we hypothesize that easterly wind directions advect air masses cooled down by surface energy*

*exchanges above ice (L304-306 in preprint manuscript) and hence refer to horizontal movement of cooled air masses, we acknowledge that it can be understood as in the sense of a vertical/convective process. We expand in L337-L340:*

For both easterly and northerly winds, air masses are advected over glacier ice or sea ice before reaching the study sites (Fig. 1c), where they can cool due to energy exchanges with the frozen surface. This site-specific meteorological effect of horizontal movement of cooled air masses may not be adequately resolved by CARRA at the local scale, which could explain the observed discrepancies to the UAV measurements.

**The authors use K-means clustering to group atmospheric observations. This is an accepted use. However, they fail to mention with what? K-means clustering is typically used in situations which highly multivariable data, though the authors only present data for temperature. My guess is that they do so on CARRA data, as they later define clusters by regional pressures. Either way, this needs to be explicitly discussed in the text before clusters can be evaluated.**

*To our assessment, we do not fail to mention with which variable we performed the clustering: We state in L131-L133 of the preprint manuscript "[...] we applied K-means clustering to categorize temperature gradients of the vertical profiles [...]". However, we acknowledge that the K-means clustering method can be presented more clearly and in the revised version we clarify which variables were clustered in L135-L146:*

Next, we applied K-means clustering to categorize the daily vertical temperature gradient profiles derived from the CARRA grid-point above tundra between June and August during the period 1991–2024, interpolated to 1 m vertical resolution. The clustering was conducted for atmospheric layers between the height up to which the troposphere is directly influenced by surface properties and up to 500 m above ground. […] The resulting clusters were then used to investigate the corresponding large-scale atmospheric conditions using ERA5 reanalysis data (Hersbach et al., 2020).

**The authors concluded that surface albedo affects the surface mass balance at their sampling site. Albedo isn't a new result and I would hope that the authors would develop a more quantitative description of albedo at their site. Also, given the weight of the importance placed on the concept of snow-albedo feedback, I would expect the word "albedo" to show up earlier in the background or methods rather than for the first time in the discussion on Line 335.**

*It is true that albedo affects surface mass balance, and as the comment states, this is not a new result; however, we feel a bit misunderstood here, as we are not sure where the Reviewer's impression arises that we state "that surface albedo affects the surface mass balance at their sampling site". What we indeed discuss in section 4.3 are effects amplifying melt due to the flat hypsometry of FIIC and refer to Davies et al. (2024) when considering ice cap albedo as a relevant*

*feedback mechanism in such a system (L346-L350 in the preprint manuscript). The snow-albedo feedback is not directly related to surface mass balance in our analysis. We wonder as to where in the manuscript this impression arose and would appreciate a clarification.*

*It is a good advice to introduce "albedo" earlier in the study and we do so in the revised version in L37-L39:*

The mass loss of the GIC in Greenland is primarily driven by surface melt, which is controlled by a complex interplay of surface energy exchanges tightly coupled with surface albedo and atmospheric conditions in the lower troposphere (Bollen et al., 2023; Noël et al., 2017; Gardner et al., 2013; Shahi et al., 2020).

**The authors have a tendency to overuse verbiage with value judgements included. Example such as Line 29 "unequivocally" or Line 61 "exceptionally" are unnecessary and not appropriately reductive to the presented and supported science in the manuscript.**

*We revised our verbiage and improved the precision when using adjectives. However, we believe that the use of "unequivocally" in the context of scientific consensus regarding the fact that anthropogenic greenhouse gas emissions are the primary cause of climate warming (L29) is relevant and appropriate. This term is also commonly used in high-level scientific compilations to make similar statements (e.g. IPCC, 2021; Siegert et al., 2025). We adapted the following other case addressing adequate use of qualitative wording:*

- *L60: Deleted "exceptionally"*

- *L265: Changed "Increasingly severe summer losses" to "increasing summer mass losses"*

- *L270: Changed "brought a sharp decline in SMB" to "brought a pronounced decline in SMB"*

**The authors tend to report relationships without quantifying them. Examples such as Line 62: What height is near surface, when is the coldest month. Line 112: What improvements? Line 129: How well?**

*Thanks for pointing this out. We follow the advice and included the following quantifications in the revised version.*

*L61-L62:*

The warmest month is July with a mean air temperature of 4 °C while the coldest month March reaches -26 °C (Gryning et al., 2021).

*L113-L116:*

CARRA is laterally forced by ERA5 reanalysis but offers a higher spatial resolution and has several improvements compared to ERA5, including enhanced assimilation of in situ observations and improved incorporation of satellite data, making it particularly suitable for studying meteorological variables at high resolution in the Arctic (Schyberg et al., 2021a; Schmidt et al., 2023).

**Point by point reply to review comment 2 (RC2)**

**Summary**

In this study, Fipper and coauthors analyze the vertical temperature structure above Flade Isblink Ice Cap (FIIC) in northeast Greenland during summer and its influence on the ice cap's surface mass balance (SMB). They first use a large number of atmospheric profiles measured by uncrewed aerial vehicles (UAVs) over difference surface types near Villum Research Station to relate the near-surface thermal gradients to surface type and assess the accuracy of the Copernicus Arctic Regional Reanalysis (CARRA) in reproducing these temperature structures. Finding a good agreement between CARRA and the UAV profiles, they use CARRA to extend the analysis in time and assess the synoptic circulation patterns that control the vertical temperature structure over FIIC. They find that the decreasing SMB trend for FIIC is related to summer atmospheric warming across most circulation patterns, and that the anticyclonic patterns that favor increased surface melt have shown an increasing trend over time.

In my assessment, this is an excellent study overall. The paper is well-written and easy to follow, with clear figures that illustrate the main findings. The use of UAV data to validate the CARRA reanalysis is innovative and the methods for this comparison appear robust. I have some comments below asking for some minor clarifications and technical corrections. Once these comments are addressed, I feel this paper will be a valuable addition to the literature on the coupling of the atmosphere and cryosphere in Greenland.

*Thanks a lot for the kind words and the overall positive assessment. This gives us courage and makes us proud.*

**Minor comments**

- General comment: Although no UAV measurements were taken at the top of the FIIC, it might be helpful to have some discussion of how applicable these results are to the ice cap in this topographically complex region. Do the authors expect that the vertical temperature profiles and gradients over the higher elevations of the ice cap would be similar to those observed at lower elevations?

*Thank you, this is an interesting point! We do not expect the vertical temperature profiles and temperature gradients at higher elevations to be similar to those at our sites, because steeper slopes compared to the sounding locations facilitate katabatic winds and in general, the near-surface climate is likely very different in higher elevations, less topographically protected environments and different surface energy exchange processes may prevail. However, we do not study this in detail and a comprehensive analysis of vertical profiles across FIIC is out of the scope of our study, which is based on local in situ profiles at VRS. However, we address this in the revised manuscript when discussing the UAV profiles (fig. 2a) in L324-L327:*

Furthermore, the profile characteristics above ice cannot be generalized to the whole ice cap, as the soundings were obtained from close to its margin. Higher elevations are topographically less sheltered, favouring stronger synoptic scale influences on the lower troposphere, and different energy exchange processes may prevail there (Oerlemans et al., 1999).

**And in using the CARRA data for the long-term analysis (e.g. L125–127), are the CARRA data only from the three fixed grid points, or are data from a larger spatial domain around FIIC examined?**

*The data for the long-term analysis are from the three fixed grid points (Fig. 1). We clarify this in the revised manuscript in L128-L130:*

We then made use of the extended time series of CARRA at the fixed grid-point coordinates for its full available period (1991–2024) for the summer months June, July and August (JJA) to assess the drivers of vertical temperature gradients at VRS.

**- L24–25: There appears to be a discrepancy between the abstract and the findings presented in Section 3.4. The abstract states that FIIC mass loss is driven by "rising summer air temperatures for all synoptic conditions", while L286–287 states that cluster 1 (low pressure) does not exhibit a significant linear trend in either summer mean SMB or summer air temperature.**

*This is an excellent point that indeed requires clarification. With the statement in the abstract, we want to point out that summer conditions are driving annual surface mass balance. While we report a Pearson correlation coefficient of -0.88 (p < 0.05) between summer air temperatures and annual SMB (L275), figure 8c demonstrates that also within cluster 1 there appears a tight relationship between summer air temperatures and summer SMB. This leads to our judgement that summer air temperatures are driving the SMB for all synoptic conditions. However, indeed the impact of different clusters on the SMB over time is a different issue. As presented in the comment, these relationships can appear contradicting. We clarify this by removing "rising" in the statement in the abstract and will state instead in L24-L25:*

Overall, mass loss of ~21 Gt occurred since 2015, driven by summer air temperatures under all synoptic conditions.

**- L75 (Fig. 1): The Fig. 1 caption states that base layers are from Google satellite imagery – is the imagery actually from a data source like Landsat or Sentinel that is indexed by Google?**

*Thanks for pointing that out. The data actually stems from Landsat/Copernicus and we use the processed product from Google. We added that to the reference and the caption for fig. 1 in L78-L79:*

The base layers in panel (a) and (c) are from Landsat satellite imagery provided by Google (2025)

**- L100: States that the temperature measurements are averaged into 12m elevation bins to account for the Imet-XQ2 sensor's vertical accuracy. Is it possible to get more accurate information on the UAV's vertical position from the UAV itself (i.e. from an onboard GPS or some other data source)?**

*This is an interesting consideration. In principle, it should be possible by analysing flightlogs from the UAVs. However, given the comparably low vertical resolution of CARRA (13 vertical levels up to 500 m including 2 m air temperature and skin temperature) and the relatively high vertical resolution of the iMet-XQ2 sensors (12 m), their accuracy is appropriate for our comparison and an enhanced vertical resolution is not expected to provide any additional insights. Therefore, we did not implement such a refinement.*

**- L131–134: Similar to my general comment above, is the K-means clustering applied only to the temperature profile data from the three fixed grid points from CARRA? Or to a more spatially extensive dataset?**

*The K-means clustering is applied to the vertical temperature gradient data from the fixed grid-point tundra as shown in fig. 1 in the manuscript. As the clustering is applied for temperature gradients above the altitude limit, up to which direct surface influences are dominant, taking the data from just one surface type is sufficient. We specify this in the manuscript to avoid confusion in L135-L137.*

Next, we applied K-means clustering to categorize the daily vertical temperature gradient profiles derived from the CARRA grid-point above tundra between June and August during the period 1991–2024, interpolated to 1 m vertical resolution.

**- ~L170 (Fig. 2): I suggest making the delta-T vertical grid line at 0 thicker, for easier visibility on the figure**

*Good suggestion for improving the readability of the figure, we adapted the figure.*

**- ~L195 (Fig. 3): This figure is really neat. I assume this figure is based on the CARRA data, and UAV data does not enter into it at all?**

*Thanks for the kind words. This is correct, it is only based on CARRA data – we specify this by adding to the figure's caption in L217:*

Both figures are only based on CARRA.

**- ~L197 and elsewhere: Are there no observations from Villum Research Station (VRS) that can be used to quantify snow cover, instead of using CARRA for snow cover? Do the studies cited in L325–327 about snow cover trends at VRS use snow cover observations from the station?**

*There are snow cover observations at VRS between 2014 and 2018 that were used by van der Schot et al. (2024). For our climatological perspective on the interplay of snow cover and air temperatures (Fig. 3) this time period is rather short, which is why we deem the use of CARRA data to be more appropriate. Furthermore, during (early) summer, snow cover typically becomes patchy. Therefore, CARRA's snow fraction data appears more suitable for studying this interplay than a point measurement at VRS. However, we compared CARRA's snow fraction data with the observed snow water equivalent and snow height and found significant correlations of $r > 0.8$ ($p < 0.05$) and an in-depth analysis of the performance of CARRA with several in-situ measurements of snow height (including VRS) is given in van der Schot et al. (2024). We added the agreement shown by van der Schot et al. (2024) in the revised manuscript in L131-L133:*

Van der Schot et al. (2024) demonstrated that CARRA performs well in representing snow cover at VRS, reporting a Pearson correlation coefficient of 0.87 between observed and CARRA-derived snow water equivalent.

**- ~L215: It would be nice to have some more discussion about what causes the varying vertical temperature gradients and temperature profiles across clusters, as shown in Fig. 4. Unless I missed it, there isn't really any physical explanation provided for why these distinct shapes of the temperature profile occur for each cluster**

*Those temperature gradients and temperature profiles emerge from clustering the gradients between 100 m and 500 m above tundra surface based on CARRA data. We show that at those altitudes synoptic conditions drive vertical temperature structures and we assign descriptive names for the clusters' synoptic conditions (Fig. 5). But indeed, we do not specifically link synoptic conditions with the vertical temperature structures and took this up in the revised manuscript. We examine the synoptic wind directions and wind speeds in 850 hPa by adding vectors of wind anomalies to fig. 5a (also see comment below). Since wind plays a key role in the mixing of air masses and the formation of temperature inversions, this addition provides a more comprehensive understanding of the processes shaping the vertical temperature structures shown in fig. 4. We address this in the discussion L377-L384:*

The synoptic regime of CL1 is characterized by the lowest $T\_850$ anomalies corresponding to northerly $W\_850$ anomalies, resulting in the lowest air temperatures and the most negative temperature gradients (< -0.5 °C 100 m$^{-1}$) between 100 m and 500 m a.g.l. at VRS (Fig. 4, Fig. 5a). In contrast, CL4 and CL5 exhibit temperature inversions in their mean vertical profiles, which appear linked to the highest $T\_850$ values and to the advection of relatively warm air masses by easterly and southerly winds. CL2 and CL3 show intermediate vertical profiles corresponding to the weakest pressure and wind anomalies in their respective synoptic regimes. These results demonstrate that above 100 m a.g.l., distinct vertical temperature structures emerge under different large-scale circulation patterns.

**- ~L237 (Fig. 5): I suggest the authors consider plotting wind vectors on the map panels of Fig. 5 (Fig. 5a), to more directly show the synoptic flow direction. For example, L364 discusses northerly wind advection during the cluster 1 regime that could be shown more directly with composite wind vectors.**

*We do agree and added vectors of wind anomalies in 850 hPa to fig. 5a accordingly in the revised version. Indeed, we see northern wind anomalies for cluster 1 and easterly/southerly wind anomalies for the other clusters, which supports our argumentation regarding the advection of air masses from certain directions at VRS in the discussion section.*

**- L245: How is SMB per unit area different from summing SMB across all grid cells on the ice cap?**

*The SMB per unit area refers to the specific surface mass balance, computed by summing the SMB over the ice cap and dividing by its area (fixed over time). This yields an area mean SMB (e.g., in m w.e.). In contrast, summing the SMB across all grid cells without dividing by area yields (including a mass conversion) the total mass change of the ice cap over a time interval (e.g., in Gt yr$^{-1}$).*

**- L304–305: Could there also be a downsloping effect from FIIC that influences the disagreement between CARRA and UAV profiles for the easterly wind regime?**

*That is a very interesting question that we considered as well. In our assessment, downslope flows from FIIC may indeed interact with synoptic easterly winds, potentially influencing the vertical structures observed around VRS and contributing to the disagreement with CARRA. This process would, however, be consistent with our hypothesis of the advection of air masses cooled by surface energy exchanges over frozen surfaces. We address this in the discussion as it highlights avenues for future research as follows in L337-342:*

For both easterly and northerly winds, air masses are advected over glacier ice or sea ice before reaching the study sites (Fig. 1c), where they can cool due to energy exchanges with the frozen surface. This site-specific meteorological effect of horizontal movement of cooled air masses may

not be adequately resolved by CARRA at the local scale, which could explain the observed discrepancies to the UAV measurements. Furthermore, katabatic winds descending from the ice cap may interact with larger-scale flows and could influence the observed discrepancies, highlighting subjects for future investigations.

**- L309–311: Also, it appears that the the overwhelming majority of the field days were categorized into Cluster 1 (low pressure) or Cluster 2 (zonal) synoptic flow, with little sampling of the anticyclonic conditions? (see Fig. 5d)**

*This is correct and a valid constraint that any campaign of a limited time period will have. The only way we can think of in order to bypass it, is to stress the reasonable agreement with CARRA and then rely on CARRA for longer-term perspectives, as we do in this study. We discuss this constraint in the revised version as follows in L387-L391:*

Analysis of the cluster occurrences (Fig. 5b–d) shows that most in situ soundings were conducted during low-pressure conditions (CL1), with only two field days falling into high-pressure regimes (CL3, CL5). This constrains the ability of the in situ profiles to capture the variability of vertical temperature structures during summer. However, validating CARRA using in situ soundings and utilising an extended CARRA time period provides a valuable way to bypass this limitation and to derive climatological insights beyond the field campaign.

**Technical corrections**

**- L250: This reference to Fig. 6c should instead be to Fig. 6b?**

*Adapted (L268).*

**- L273: "rare" --> "rarely"**

*Adapted (L291).*

**- L349–350: Something about the grammar is off in this sentence... should this be written instead as "...pronounced as \*a\* disproportional drop..."?**

*Adapted (L404).*

**- L380: "Why not also CL1 shows" --> "why CL1 does not also show"**

*Adapted (L434).*

**References**

Davies, B., McNabb, R., Bendle, J., Carrivick, J., Ely, J., Holt, T., Markle, B., McNeil, C., Nicholson, L., and Pelto, M.: Accelerating glacier volume loss on Juneau Icefield driven by hypsometry

and melt-accelerating feedbacks, Nat Commun, 15, 5099, https://doi.org/10.1038/s41467-024-49269-y, 2024.

IPCC: Framing, Context, and Methods, in: Climate Change 2021 – The Physical Science Basis: Contribution of Working Group I to the Sixth Assessment Report of the Intergovernmental Panel on Climate Change, edited by: IPCC, Cambridge University Press, 147–286, https://doi.org/10.1017/9781009157896.003, 2021.

Siegert, M., Sevestre, H., Bentley, M. J., Brigham-Grette, J., Burgess, H., Buzzard, S., Cavitte, M., Chown, S. L., Colleoni, F., DeConto, R. M., Fricker, H. A., Gasson, E., Grant, S. M., Gulisano, A. M., Hancock, S., Hendry, K. R., Henley, S. F., Hock, R., Hughes, K. A., Karentz, D., Kirkham, J. D., Kulessa, B., Larter, R. D., Mackintosh, A., Masson-Delmotte, V., McCormack, F. S., Millman, H., Mottram, R., Moon, T. A., Naish, T., Nath, C., Orlove, B., Pearson, P., Rogelj, J., Rumble, J., Seabrook, S., Silvano, A., Sommerkorn, M., Stearns, L. A., Stokes, C. R., Stroeve, J., and Truffer, M.: Safeguarding the polar regions from dangerous geoengineering: a critical assessment of proposed concepts and future prospects, Front Sci, 3, https://doi.org/10.3389/fsci.2025.1527393, 2025.

van der Schot, J., Abermann, J., Silva, T., Rasmussen, K., Winkler, M., Langley, K., and Schöner, W.: Seasonal snow cover indicators in coastal Greenland from in situ observations, a climate model, and reanalysis, The Cryosphere, 18, 5803–5823, https://doi.org/10.5194/tc-18-5803-2024, 2024.